# The emergence of multiple retinal cell types through efficient coding of natural movies

**Samuel A. Ocko**[*][†]**, Jack Lindsey**[*]**, Surya Ganguli**[1]**, Stephane Deny**[†]
Department of Applied Physics, Stanford and [1]Google Brain, Mountain View, CA

## Abstract

One of the most striking aspects of early visual processing in the retina is the immediate parcellation of visual information into multiple parallel pathways, formed by different retinal ganglion cell types each tiling the entire visual field. Existing theories of efficient coding have been unable to account for the functional advantages of such cell-type diversity in encoding natural scenes. Here we go beyond previous theories to analyze how a simple linear retinal encoding model with different convolutional cell types efficiently encodes naturalistic spatiotemporal movies given a fixed firing rate budget. We find that optimizing the receptive fields and cell densities of two cell types makes them match the properties of the two main cell types in the primate retina, midget and parasol cells, in terms of spatial and temporal sensitivity, cell spacing, and their relative ratio. Moreover, our theory gives a precise account of how the ratio of midget to parasol cells decreases with retinal eccentricity. Also, we train a nonlinear encoding model with a rectifying nonlinearity to efficiently encode naturalistic movies, and again find emergent receptive fields resembling those of midget and parasol cells that are now further subdivided into ON and OFF types. Thus our work provides a theoretical justification, based on the efficient coding of natural movies, for the existence of the four most dominant cell types in the primate retina that together comprise 70% of all ganglion cells.

## 1 Introduction

The time honored principle that the visual system evolved to efficiently encode the structure of our visual world opens up the tantalizing possibility that we can predict, *ab initio*, the functional organization of visual circuitry simply in terms of the statistical structure of natural scenes. Indeed, efficient coding theory has achieved several successes in the retina by simply considering coding of static spatial scenes [1, 2, 3] or mostly temporal sequences [4]. However, such theories have not yet accounted for one of the most salient aspects of retinal computation, namely the existence of a diversity of retinal ganglion cell types, each forming a mosaic that uniformly tiles the visual field [5].

A few theoretical studies have suggested reasons for different cell types. One suggestion is the feature detector hypothesis [6, 7, 8], or the need to detect highly specialized, behaviorally relevant environmental cues. However, many cell types respond broadly to general classes of stimuli whose *direct* behavioral relevance remains unclear [9, 10, 11]. Another line of argument involves metabolic efficiency. In particular the division of ganglion cells into rectifying ON and OFF populations is more metabolically efficient than linear encoding with a single population [12], and the asymmetry between ON and OFF cells can be related to the asymmetric distribution of light intensity in natural spatial scenes [13]. Another efficient coding argument explains why two populations with similar receptive fields (RFs) have different activation thresholds in the salamander retina [14].

---

[*]Equal contribution. All code available at https://github.com/ganguli-lab/RetinalCellTypes.
[†]Corresponding authors: samocko@gmail.com and stephane.deny.pro@gmail.com.

Here we go beyond previous efficient coding theories of the retina by optimizing convolutional retinal models with multiple cell types of differing spatial densities to efficiently encode the *spatiotemporal* structure of natural *movies*, rather than simply the spatial structure in natural scenes. Indeed, psychophysical studies of human sensitivity [15, 16] suggest our visual system is optimized to process the spatiotemporal information content of natural movies. Our theory enables us to account for several detailed aspects of retinal function. In particular, the primate retina is dominated by four types of ganglion cells, ON midget and parasol cells and their OFF counterparts. Together, these types constitute 68% of all ganglion cells [17], and more than 95% in the central retina [18]. Midget cells are characterized by (1) a high density of cells (52% of the whole population), (2) a small spatial RF, (3) slow temporal filtering, and (4) low sensitivity – as measured by the slope of their contrast-response function. In contrast, parasol cells are characterized by (1) a low density of cells (16% of the whole population), (2) a large RF, (3) fast temporal filtering and (4) high sensitivity [19, 20, 21, 22]. Moreover, the density ratio of midget to parasol cells systematically decreases across retinal eccentricity from the fovea to the periphery [23].

Remarkably, our theory reveals how all these detailed retinal properties arise as a natural consequence of the statistical structure of natural movies and realistic energy constraints. In particular, our theory simultaneously accounts for: (1) why it is beneficial to have these multiple cell types in the first place, (2) why the four properties of cell density, spatial RF-size, temporal filtering speed, and contrast sensitivity co-vary the way they do across midget and parasol types, and (3) quantitatively explains the variation in midget to parasol density ratios over retinal eccentricities. Moreover, our theory, combined with simulations of efficient nonlinear encoding models, also accounts for the existence of both ON and OFF midget and parasol cells. Thus simply by extending efficient coding theory to multiple cell types and natural movies with a realistic energy constraint, we account for cell-type diversity that captures 70% of all ganglion cells.

## 2    A theoretical framework for optimal retinal function

**Retinal model.** We define a ganglion cell type as a convolutional array of neurons sampling linearly from a regularly spaced array of $N_p$ photoreceptors, indexed by $i = 0, \ldots, N_p - 1$ (Fig. 1A). We model photoreceptors as linearly encoding local image contrast. We also assume the $N_{\mathcal{C}}$ ganglion cells of cell type $\mathcal{C}$, indexed by $j = 0, \ldots, N_{\mathcal{C}} - 1$, each have a common spatiotemporal RF, defined as $F_{\mathcal{C}}(i, \Delta t)$, whose center is shifted to a position $j \cdot s_{\mathcal{C}}$ on the photoreceptor array. Here $s_{\mathcal{C}}$ is the *convolutional stride* of type $\mathcal{C}$, which is an integer denoting the number of photoreceptors separating adjacent RF centers of ganglion cells of type $\mathcal{C}$, so that $N_p = s_{\mathcal{C}} N_{\mathcal{C}}$. This yields a retinal model

$$Y_{\mathcal{C},j}(t) = \sum_{i=0}^{N_p-1} \sum_{t'=0}^{T-1} F_{\mathcal{C}}(-i + [j \cdot s_{\mathcal{C}}], t - t') \cdot X_i(t') + \eta_{\mathcal{C},j}(t), \tag{1}$$

where $X_i(t')$ is activation of photoreceptor $i$ and time $t'$, $\eta_{C,j}(t)$ is additive noise that is white across both space ($\mathcal{C}$ and $j$) and time ($t$) with constant variance $\sigma_{\eta}^2$ [24], and $Y_{\mathcal{C},j}(t)$ is the firing rate of ganglion cell $j$ of type $\mathcal{C}$. We work in one spatial dimension (generalizing to two is straightforward). This linear model will provide conceptual insight into spatiotemporal RFs of different cell types through exact mathematical analysis. However, in Sec. 5 we also consider a nonlinear version of the model to account for rectifying properties of different cell types.

**Natural movie statistics.** We approximate natural movies by their second order statistics (Fig. 1B), assuming Gaussianity. Natural movies are statistically translation invariant, and so their second order statistics are described by their Fourier power spectrum, which follows an approximately space-time separable power law as function of spatial ($k$) and temporal ($\omega$) frequency [25, 26, 27]:

$$S(k, \omega) \propto 1/|k|^2 |\omega|^2. \tag{2}$$

**Optimization Framework.** We assume the objective of the retina is to faithfully encode natural movies while minimizing overall ganglion cell firing rate. We quantify encoding fidelity by the amount of input variance $\mathcal{V}$ explained by the optimal minimum mean squared error (MMSE) reconstruction of photoreceptor activity patterns from ganglion cell outputs by a linear decoder. Moreover, we assume an overall penalty on output firing rate that is proportional to a power $p$ of the rate. This yields an objective function to be maximized over the set of convolutional filters $\{F_{\mathcal{C}}\}$:

$$\mathcal{O}(\{F_{\mathcal{C}}\}) = \mathcal{V} - \lambda \sum_{j=0}^{N_C-1} \langle Y_j^2 \rangle^{p/2}. \tag{3}$$

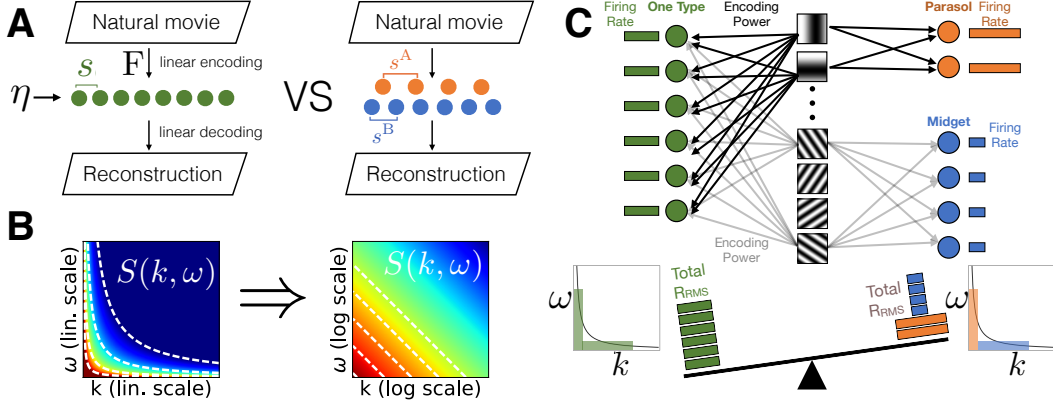

Figure 1: An efficient coding model for multiple convolutional cell types. A) Natural movies are encoded in a set of ganglion cells (circles) through linear filters corrupted by noise (Eq. 1). Left: one convolutional cell type with stride $s$ (green). Right: two different cell types (blue and orange) with different strides $s^{\mathbb{A}}$ and $s^{\mathbb{B}}$. B) The Fourier power spectrum $S(k, \omega)$ of natural movies decays as a separable power law in both spatial ($k$) and temporal ($\omega$) frequency (see Eq. 2). Dashed lines are iso-contours of constant power in linear (left) and logarithmic (right) axes with power varying from high (red) to low (blue). Note that, aside from the origin, the most powerful Fourier modes are contained in two distinct regions: (1) low $k$, high $\omega$, and (2) high $k$, low $\omega$. C) We will show that two convolutional cell types (orange, blue) encode visual information more efficiently than one type (green) by specializing their filters to cover these two regions. The orange, parasol-like cell type specializes to region (1) using a small number of cells at large stride with large spatial RFs (low $k$), and fast temporal filters (high $\omega$) that fire sensitively at high rates. The blue, midget-like cell type specializes to region (2) using a large number of cells at small stride with small spatial RFs (high $k$) and slow temporal filters (low $\omega$) using low firing rates. Together, these two specialized cell types can encode photoreceptor input patterns with the same fidelity as a single, undifferentiated cell type (green), with less total firing rate.

Here $\lambda$ is a parameter that trades off between the competing desiderata of maximizing encoding fidelity versus minimizing firing rate. Our final results will focus on the choice $p = 1$, motivated by the linear relationship between metabolic cost and firing rate [28, 29]. This choice is similar to an $\ell_1$ penalty used in [30, 31, 2]. However we also consider more general $p$, including $p = 2$, both to connect to prior work on efficient coding [1, 3] and as a building block for solving the $p = 1$ case.

**Outline.** In Sec. 3, we prove mathematically that multiple cell types enable a more efficient code (i.e. same or better encoding fidelity with lower firing rates) than a single cell type, as long as $p < 2$. The fundamental idea is that different cell types allow higher efficiency by *specializing* to different regions of the power spectrum of natural movies (Fig. 1C). In Sec. 4, we find the optimal cell types for natural movies, demonstrating that the best two-type strategy substantially out-performs the best one-type strategy. We then compare these optimal types to midget and parasol cells in the primate retina and find striking agreement between optimal and biological cell types. In Sec. 5, we extend our theory to non-linear ganglion cells and account for both ON and OFF midget and parasol cells.

## 3   Mathematical proof of the benefit of multiple cell types

Here we derive mathematically how multiple specialized cell types can confer an efficient coding advantage compared to a single cell type (Fig. 1A,C). In Sec. 3.1, we start with the simple case of a single cell type with stride 1 yielding an equal number of ganglion cells and photoreceptors, encoding static images [1, 32]. We then extend this framework to varying strides (Sec. 3.2), encoding natural movies (Sec. 3.3), and multiple cell types, proving that they can confer an advantage (Sec. 3.4). We will solve for the optimal RFs and strides of each cell type in Sec. 4.

### 3.1   Encoding $N_p$ photoreceptors with $N_C = N_p$ ganglion cells of a single type

As a warmup, for purely spatial scenes, we first consider optimizing the single cell-type retinal filter $F_C(i, \Delta t)$ in Eq. 1 under the objective function in Eq. 3, in the simple case of stride $s_C = 1$ so that $N_C = N_p$. Since we only have one cell type, we drop the cell-type index $C$ in the following. The case

of $N_\mathcal{C} = N_p$ simplifies because we can ignore aliasing [33], which we address in the next section. Thus we can show (App. A) that each spatial Fourier mode $\tilde{X}(n)$ of photoreceptor patterns maps in one-to-one fashion onto a single spatial Fourier mode $\tilde{Y}(m)$ of ganglion cell patterns (Fig. 2A1):

$$Y_j = \sum_{i=0}^{N_p-1} F(-i+j) \cdot X_i + \eta_j \qquad \Rightarrow \qquad \tilde{Y}(m) = \delta_{m,n}\tilde{F}(n) \cdot \tilde{X}(n) + \tilde{\eta}(m), \qquad (4)$$

where $\delta$ is the Kronecker delta function, $n \in \{-N_p/2 + 1, \ldots 0, 1, \ldots N_p/2\}$ indexes photoreceptor Fourier modes, $m \in \{-N_\mathcal{C}/2 + 1, \ldots 0, 1, \ldots N_\mathcal{C}/2\}$ indexes ganglion cell Fourier modes, and $\tilde{\eta}(m)$ is the spatial Fourier transform of the noise (which also has variance $\sigma_\eta^2$). $\tilde{F}$ is the Fourier transform of F across photoreceptors, rescaled by $\sqrt{N_\mathcal{C}}$. Each mode number $n$ ($m$) corresponds to a photoreceptor (ganglion cell) spatial frequency $k_n \equiv 2\pi n/N_p$ ($p_m \equiv 2\pi m/N_\mathcal{C}$). The power $S(n)$ in photoreceptor mode $n$ is simply proportional to the power $S(k, \omega)$ in natural movies (Eq. 2) evaluated at spatial frequency $k = k_{|n|}$. Finally, because image statistics are translation invariant, the objective (Eq. 3) can be written (App. A) in terms of independent photoreceptor spatial modes (here $p = 2$):

$$\mathcal{O} = \mathcal{V} - \lambda \sum_{j=0}^{N_\mathcal{C}-1} \langle|Y_j|^2\rangle = \sum_{n=-N_p/2+1}^{N_p/2} \left( \frac{|\tilde{F}(n)|^2 S(n)^2}{\sigma_\eta^2 + |\tilde{F}(n)|^2 S(n)} - \lambda \big[|\tilde{F}(n)|^2 S(n) + \sigma_\eta^2 \big] \right). \quad (5)$$

Thus $\mathcal{O}$ can be maximized independently for each filter mode $n$, yielding the optimal filter (App. A):

$$|\tilde{F}_{\text{Opt}}(n)|^2 = Q^{-1}(n)\big[\mathcal{H} - Q^{-1}(n)\big]_+, \text{ where } Q(n) = \sqrt{S(n)/\sigma_\eta^2}, \; \mathcal{H} = 1/\sqrt{\lambda}, \qquad (6)$$

where $Q(n)$ is a measure of the *quality* of photoreceptor Fourier mode, or input channel $n$.

This solution has an appealing water-filling [34] interpretation (Fig. 2A2) in which each channel $n$ of quality $Q(n)$ corresponds to a beaker with base height and width both equal to $Q(n)^{-1}$. These beakers are filled with water up to height $\mathcal{H} = 1/\sqrt{\lambda}$, and the power $|\tilde{F}_{\text{Opt}}(n)|^2$ assigned to filter mode $n$ is simply the *volume* of water in beaker $n$. Thus extremely low quality channels with beaker base $Q(n)^{-1}$ greater than the water height $\mathcal{H}$ are not used. Similarly, high quality channels with a low base are not assigned much filter strength because they are also narrow. Thus the optimal solution assigns filter strength as a non-monotonic function of channel quality (Fig. 2A3), favoring channels of intermediate quality, eschewing extremely low quality channels that do not contribute much to encoding fidelity, while attenuating channels that are already high-quality whose amplification would yield a cost in firing rate that outweighs the improved coding fidelity. As the penalty $\lambda$ in firing rate is reduced, the water height $\mathcal{H}$ increases, and more lower quality channels are used by the optimal filter.

Because the power spectrum of natural movies decays with spatial frequency [25], higher (lower) quality channels correspond to lower (higher) spatial frequencies. Thus the non-monotonic optimal filter strength as a function of channel quality (Fig. 2A3) leads to two qualitative effects (Fig. 2A4): (1) the attenuation of very low frequency high quality channels relative to intermediate frequency channels (spatial whitening) driven primarily by the need to lower firing rate, and (2) the eschewing of very high frequency low quality channels (spatial smoothing), which do not contribute strongly to encoding fidelity.

### 3.2 Encoding $N_p$ photoreceptors with $N_\mathcal{C} < N_p$ ganglion cells of a single type

In the case of strides greater than 1, with fewer ganglion cells than photoreceptors, more than one spatial Fourier mode of photoreceptor activity can map to the same spatial Fourier mode of ganglion cell activity, a phenomenon known as *aliasing*. Indeed, not only does photoreceptor mode index $m$ map to ganglion cell mode index $m$, as in Sec 3.2, but so does every other photoreceptor mode $n$ separated from $m$ by an integer multiple of $N_\mathcal{C}$ (Fig. 2B, App. B), yielding the map

$$Y_j = \sum_i F(-i+js) \cdot X_i + \eta_j \Rightarrow \tilde{Y}(m) = \sum_{n=m+\text{n'}N_\mathcal{C}} \tilde{F}(n) \cdot \tilde{X}(n) + \tilde{\eta}(m), \qquad (7)$$

where n' ranges over integers such that $n = m + \text{n'}N_\mathcal{C}$ enumerates all photoreceptor frequencies $n$ within the bounds $-N_p/2 < n \le N_p/2$ which alias to the same ganglion cell frequency $m$. Despite the many-to-one map from photoreceptor to ganglion cell Fourier modes, one can still optimize Eq. 3 independently over different filter modes akin to Eq. 5 through the following argument. The firing of each ganglion cell frequency $m$ comes from the set of photoreceptor frequencies which alias to it, i.e. $n = m + \text{n'}N_\mathcal{C}$ (Fig. 2B1). First we show that it is optimal for a single ganglion cell mode $n$ to draw

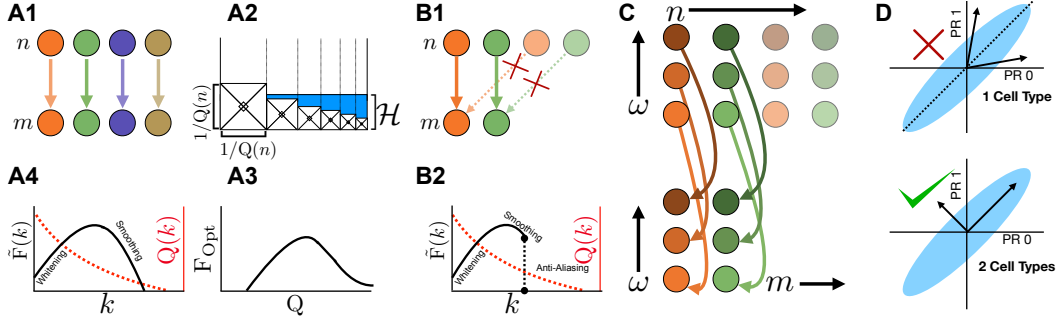

Figure 2: A1) In the convolutional framework of [1], every photoreceptor spatial frequency (upper dots) maps to a corresponding ganglion cell spatial frequency (lower dots). A2) The optimal filter strength assigned to each mode $n$ can be viewed as the volume of water assigned to a corresponding beaker whose base height and width are inversely related to channel quality $Q(n)$, and the water height $\mathcal{H}$ is inversely related to the firing rate penalty parameter $\lambda$. A3) The optimal filter strength is a non-monotonic function of channel quality. A4) This also leads to non-monotonic optimal filter strength as a function of spatial frequency. B1) With fewer ganglion cells than photoreceptors, the optimal filter will sample *only* from the lowest photoreceptor spatial frequencies that map in one-to-one fashion to the lowest ganglion cell spatial frequencies, and will ignore the higher photoreceptor frequencies that alias to the same ganglion cell frequencies (App. B). B2) Thus the optimal filter achieves a similar solution as in A4, with an additional upper bound on frequency to avoid aliasing. C) In spacetime, the optimal filter maps spatiotemporal photoreceptor frequencies (upper dots) to ganglion cell frequencies (lower dots) in a one-to-one fashion, ignoring higher photoreceptor spatial frequencies to avoid spatial aliasing. Different spatial (temporal) frequencies are indicated by shade (color). D) (See Sec. 3.4) The blue ellipsoid illustrates a correlated stimulus covariance across two photoreceptors (PR). Top: the two arrows denote two filters $\vec{F}_0$ and $\vec{F}_1$ of two ganglion cells of a single cell type, related by a convolutional translation, modulo $N_p = 2$ and therefore reflection-symmetric about the diagonal (Sec. 3.4). Bottom: the two arrows denote two rotated filters $\vec{F}_A$ and $\vec{F}_B$ that each specialize to a different eigenbasis vector of the stimulus covariance, thereby differentiating into two cell types, enabling a more efficient neural code with the same fidelity but lower firing rate cost.

*only* from the input eigen-mode with largest power (App. B.1). Now given the lowest photoreceptor spatial frequencies have the highest power, the optimal convolutional filter should sample only from the lowest photoreceptor frequencies, and not from the higher aliasing frequencies. Thus the optimal filter has $\tilde{F}(n) = 0$ for all $n$ with $|n| > N_C/2$. Therefore, the optimal filter in frequency space is simply a scaled, truncated version of Eq. 6 (Fig. 2 B1, B2):

$$|\tilde{F}_{\text{Opt}}(n)|^2 = H\left(N_C/2 - |n|\right) Q^{-1}(n) \left[\mathcal{H} - Q^{-1}(n)\right]_+, \ \text{ where } \ Q(n) = \sqrt{S(n)/\sigma_\eta^2}, \ \mathcal{H} = 1/\sqrt{\lambda}, \quad (8)$$

where H is the Heaviside function.

Note the optimal upper frequency cutoff to avoid aliasing naturally yields tiling, in which the spatial RF width becomes proportional to the stride [33]. To see this, consider a cell type with a stride in physical space of length $s$. Spatial frequencies higher than $O(1/s)$ will lead to aliasing, yielding a frequency cut-off $k_s \propto 1/s$. Further assume, for simplicity, that the water-filling solution fills all spatial frequency modes up to $k_s$ with the same amplitude, and chooses the same phase. This yields a box Fourier spectrum whose inverse spatial RF is a *sinc* function whose first zero crossing occurs at spatial scale $1/k_s \propto s$. Thus the RF width is proportional to stride, and cells with high (low) frequency cut-offs have small (large) RF widths and strides.

### 3.3 Generalizing the framework to spatiotemporal movies for a single cell type

With the addition of time, we can Fourier transform Eq. 1 in both space and time, yielding

$$\tilde{Y}(m,\omega) = \sum_{n=m+n'N_C} \tilde{F}(n,\omega) \cdot \tilde{X}(n,\omega) + \tilde{\eta}(m,\omega), \quad (9)$$

where $X_i(t')$ is the activity of photoreceptor $i$ at time $t'$, $\tilde{F}$ is the Fourier transform of F rescaled by $\sqrt{N_C T}$, and $\tilde{\eta}(m,\omega)$ is the Fourier transform of the noise. Note that with fewer ganglion cells

than photoreceptors, there will be a many-to-one map from photoreceptor spatial frequency to each ganglion cell spatial frequency as in Eq. 7, but a one-to-one map from photoreceptor to ganglion cell temporal frequencies (Fig. 2C, App. C). As in Sec. 3.2, the optimal filter will map the lowest photoreceptor spatial frequencies one-to-one to the lowest ganglion cell frequencies, while ignoring higher photoreceptor spatial frequencies to avoid aliasing. Moreover, within this aliasing constraint, spatiotemporal photoreceptor frequencies map one-to-one to ganglion cell frequencies, yielding an optimal solution given by Eq. 8 with channel quality depending on spacetime power $\mathbf{S}(\mathbf{k}_n, \omega)$.

### 3.4 Interplay of firing rate penalty and the benefit of multiple cell types

To build intuition for when and why multiple cell types can enable more efficient neural codes, we consider the simplest possible scenario of $N_p = 2$ photoreceptors and two convolution ganglion cells of a single type. The stimulus statistics and ganglion cell filters are given by (see also Fig. 2D):

$$\mathbf{C}_{XX} = \begin{pmatrix} 1 & c \\ c & 1 \end{pmatrix}, \quad \vec{F}_0 = \begin{pmatrix} f_0 \\ f_1 \end{pmatrix}, \quad \vec{F}_1 = \begin{pmatrix} f_1 \\ f_0 \end{pmatrix}.$$

Note the two filters are equal up to a translation (modulo $N_p$) and therefore obey the convolutional constraint. Let's call $\mathbf{D}$ the optimal decoder. Then the reconstruction $\vec{X}_r$ of the input is:

$$\vec{X}_r = \mathbf{D}(\mathbf{F}\vec{X} + \vec{\eta}) = \mathbf{D}\mathbf{F}\vec{X} + \mathbf{D}\vec{\eta}.$$

Here $\mathbf{F}$ is a 2 by 2 filter matrix whose rows are given by the two ganglion cell filters. Now the decoding performance is unaffected by an orthogonal rotation of the rows of $\mathbf{F}$. Indeed, when $\mathbf{F} \to \mathbf{R}\mathbf{F}$, we can transform $\mathbf{D} \to \mathbf{D}\mathbf{R}^{-1}$ yielding the reconstruction

$$\vec{X}_r = \mathbf{D}\mathbf{R}^{-1}(\mathbf{R}\mathbf{F}\vec{X} + \vec{\eta}) = \mathbf{D}\mathbf{F}\vec{X} + \mathbf{D}\mathbf{R}^{-1}\vec{\eta}.$$

Because $\mathbf{R}$ is a rotation (i.e. $\mathbf{R}^{-1} = \mathbf{R}^T$) and $\vec{\eta}$ is isotropic Gaussian white noise, the statistics of $\vec{X}_r$ conditioned on $\vec{X}$, and thus the explained variance $\mathcal{V}$, is unchanged by the rotation. More formally, the explained variance can be computed to be $\mathcal{V} = 2\mathrm{Tr}\,\mathbf{D}\mathbf{F}\mathbf{C}_{XX} - \mathrm{Tr}\,\mathbf{D}\mathbf{F}\mathbf{C}_{XX}(\mathbf{D}\mathbf{F})^T - \sigma_\eta^2\mathrm{Tr}\mathbf{D}\mathbf{D}^T$ [35], and is independent of the transformation effected by $\mathbf{R}$. This yields an entire manifold of optimal filter matrices $\mathbf{F}$ with the *same* explained variance $\mathcal{V}$.

Now consider a *particular* choice of rotation $\mathbf{R}$ that rotates the two convolutional filters into the eigenbasis of $\mathbf{C}_{XX}$:

$$\vec{F}_A = \frac{\vec{F}_0 + \vec{F}_1}{\sqrt{2}} = \frac{(f_0 + f_1)}{\sqrt{2}}\begin{pmatrix} 1 \\ 1 \end{pmatrix}, \vec{F}_B = \frac{\vec{F}_0 - \vec{F}_1}{\sqrt{2}} = \frac{(f_0 - f_1)}{\sqrt{2}}\begin{pmatrix} 1 \\ -1 \end{pmatrix}.$$

The rotated filters $\vec{F}_A$ and $\vec{F}_B$ are no longer related by any translation. Thus the convolutional constraint is relaxed and they are analogous to two different cell types. We now compare the signal component of the total firing rate cost for the single cell-type convolutional filters, given by:

$$\langle Y_0^2 \rangle^{p/2} + \langle Y_1^2 \rangle^{p/2} = 2\left(f_0^2 + f_1^2 + 2cf_0f_1\right)^{p/2}, \tag{10}$$

with the rotated, specialized, two-cell type filters, given by

$$\langle Y_{\mathbb{A}}^2 \rangle^{p/2} + \langle Y_{\mathbb{B}}^2 \rangle^{p/2} = \left((1+c)(f_0 + f_1)^2\right)^{p/2} + \left((1-c)(f_0 - f_1)^2\right)^{p/2}. \tag{11}$$

As long as $c \neq 0$ and $p < 2$, the rotated (Eq. 11) two-type solution uses a lower firing rate budget than the one-type solution (Eq. 10). We generalize this proof in App. D to arbitrary numbers of cells, convolutional types and natural movie statistics. Thus intriguingly we find a sharp transition in the exponent $p$ relating firing rate to cost, with multiple cell types favored if and only if $p < 2$.

Some prior work on efficient coding [1, 3] employed an $\ell_2$ penalty on firing rate (i.e. $p = 2$), while others [30, 31, 2] have employed an $\ell_1$ penalty (i.e. $p = 1$ in our Gaussian scenario). We note that energetic considerations suggest that metabolic cost is linearly related to firing rate [28, 29] (i.e. $p = 1$). Prior knowledge that multiple retinal cell types do indeed exist, in addition to these energetic considerations, lead us to consider $p = 1$ in the following, corresponding to a penalty on the root-mean-squared (RMS) firing rates, summed over all cells.

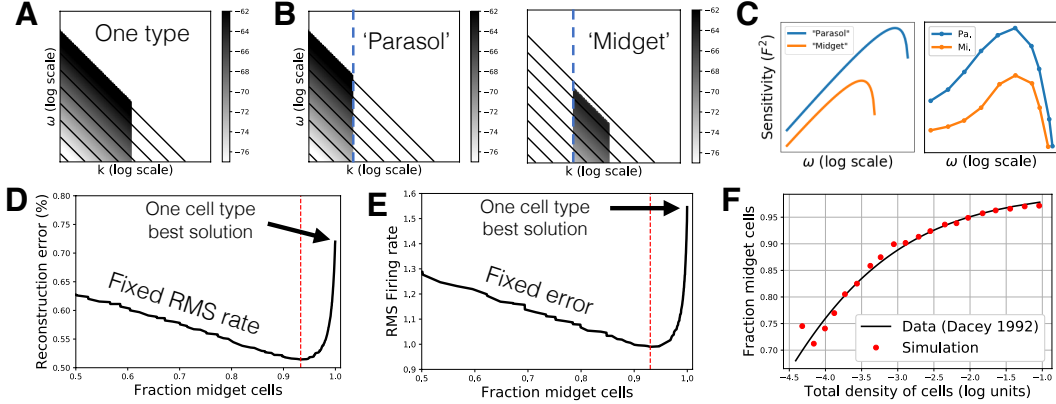

Figure 3: Optimal cell types match properties of midget and parasol cells. A) Optimal RF power spectra for a single cell type with darker shades denoting higher filter strength. Black lines are iso-contours of the power spectrum of natural movies. B) Optimal RF power spectra for two cell types with same conventions as A. C) Left: RF power spectrum of the two cell types along the frequency axis at fixed spatial frequency (the blue dashed lines in B). Right: Measured sensitivity or contrast gain as a function of temporal frequency of real midget and parasol cells (reprinted from [36]). D) Reconstruction error as a function of the fraction of midget cells for the RMS firing rate budget at which the optimal fraction is 93% (red line), consistent with the fraction found at a certain retinal eccentricity [18]. E) RMS firing rate budget as a function of the fraction of midget cells, for a fixed reconstruction error. Note the optimal one type solution (last point on the right) requires a 50% higher firing rate than the optimal two-type solution. F) Fraction of midget cells as a function of the total density of cells. Red points: optimal fractions predicted by theory (only one parameter was fitted to the data, see text). Line: fit to the fraction of midget across the human retina estimated from [23].

## 4 Comparison of theoretically derived cell types to primate retinal cell types

Here we optimize the efficient coding objective in Eq. 3, with a metabolically motivated firing rate penalty corresponding to $p = 1$, using the procedure described in App. E. In particular, for the same total RMS firing rate budget (obtained by increasing $\lambda$ in Eq. 3 until we first match this budget) we found both the optimal single (Fig. 3A) and two (Fig. 3B) cell type solutions. For the two-cell type solutions, we additionally scanned the two strides of each cell type. Given that for each cell type $\mathcal{C}$, the number of cells $N_{\mathcal{C}}$ and stride $s_{\mathcal{C}}$ are related by $s_{\mathcal{C}} N_{\mathcal{C}} = N_p$, varying the two strides is equivalent to varying the fractions of each cell type. All such two-cell type solutions had the same RMS rate, but varying encoding fidelity (the variance explained term $\mathcal{V}$ in Eq. 3). Thus for each firing rate budget, we find an optimal cell-type ratio with highest encoding fidelity. In Fig. 3AB we employed a budget that yielded an optimal cell-type ratio matching that of midget to parasol cells in the primate retina [18]. However, the general structure of the resultant RF power spectra in Fig. 3AB is robust to the choice of total RMS firing rate budget.

Remarkably, this general structure of the theoretically derived two-cell-type solution matches many properties of biologically observed primate retinal cell types. The first type corresponds to parasol cells, covering low spatial frequencies (implying large spatial RFs with large stride and low number density) and high temporal frequencies (implying fast temporal filtering). The second type corresponds to midget cells, covering a large number of high spatial frequencies (implying small spatial RFs with small stride and high number density) and low temporal frequencies (implying slow temporal filtering). Moreover, a single slice of the RF power spectrum along temporal frequency at a fixed spatial frequency (Fig. 3C, left) reveals that our theoretically derived "parasol" cell type has higher sensitivity (i.e. filter strength, or gain between input-contrast and output response) compared to the "midget" cell type, consistent with observations from the primate retina (Fig. 3C, right [36]). Thus as promised in Fig. 1C, the striking covariation of the four distinct features (cell density, spatial RF-size, temporal filtering speed, and contrast sensitivity) across the two dominant primate retinal cell types, arises as a simple emergent property of the two tailed structure of the natural movie power spectrum. By specializing to these two tails, the two-cell type solution in Fig. 3B can achieve *higher* encoding fidelity at the *same* RMS firing rate budget compared to the single-cell type solution in Fig. 3A. Indeed for the common firing rate budget chosen in Fig. 3AB, the optimal two-cell type solution achieves a 34% reduction in reconstruction error compared to the single type solution (Fig.

3D). Conversely, at a fixed reconstruction error (of 0.5%), two cell types are 33% more efficient than one in terms of total RMS firing rate (Fig. 3E). More generally, across for any non-zero firing rate budget the two-type solution achieves higher encoding fidelity, and for any desired encoding fidelity, the two-type solution requires lower firing rates.

The fixed budget shown in Fig. 3D and the fixed reconstruction error shown in Fig. 3E, were chosen such that optimal fractions of midget and parasol cells were 93% and 7%, respectively, consistent with those found at certain eccentricities of the primate retina [23]. However, in our model the optimal fractions change as the firing rate budget is increased (or equivalently, as the reconstruction error is decreased). The total density of cells in the optimal solution computed by the model also varies with the firing rate budget. Thus, our model makes a specific numerical prediction relating total cell density to the ratio of midget to parasol cells. The total density of cells varies across eccentricity by 3 orders of magnitude in the primate retina. In Fig. 3F, we plot the predicted evolution of the percentage of midget cells with cell density and compare it to the evolution of this percentage estimated from biological data [23] (see App. F for estimation method). Our model involves only one adjustable parameter to account for our arbitrary choice of units of cell density. Remarkably, we find an excellent match between theory and experiment in Fig. 3F, providing further evidence that the principle of efficient encoding of natural movies under a limited firing rate budget may be driving the functional organization of the primate retina.

## 5 A neural network simulation for linear-non-linear neurons

While the linear theory accounts for several properties of midget and parasol cells, it suffers from two main deficiencies. First, like previous efficient coding theories [1, 32], it only predicts the power of RF Fourier spectra, leaving the phase, and therefore the full spacetime RF unspecified. Second, it cannot account for rectifying nonlinearities, leading to the partition of ganglion cells into ON and OFF types. Here we remedy these deficiencies through neural network simulations, in which we nonlinearly autoencode natural movies with *two* spatial dimensions and one temporal dimension using three-dimensional convolutional neurons (full simulation details are given in App. G).

The main simulation ingredients include: (1) enforcing nonnegativity of neural firing rates through a ReLU nonlinearity in the ganglion cell encoding layer, (2) an $\ell_2$ penalty on total weight magnitude, corresponding to a cost for synaptic connections [37], (3) encouraging decoding input stimuli with a short but non-zero temporal lag [38, 39], (4) implementing a firing rate budget with an $\ell_1$ penalty on total firing rate. We assume four cell types, and optimize the number of cells allotted to each type. To match the fact that our image contrast distribution is zero mean Gaussian, and therefore symmetric about the origin, we pair the types into two pairs of equal-size populations and keep the number of cells the same across each pair during this optimization, expecting that ON and OFF homologous types will emerge. It would be interesting to explore skewed image statistics and test whether these would yield ON-OFF asymmetries, as are found in biological retinas [13]. We confirmed that our equal pairing of types is a locally optimal cell type allocation (for our symmetric image statistics) by performing a stability analysis around the best paired solution found (see App. G).

We optimize the number of neurons allocated to each type by grid search and their corresponding RFs by gradient descent (Fig. 4, App. G). The four optimal cell type RFs are strikingly similar to those of real ON-OFF midget and parasol cells found in the primate retina (see Fig. 4A-D for representative examples of near-optimal neural network RFs, Fig. 4E-H for macaque data, also see App. I). Both biphasic temporal filters and the characteristic center-surround RF shape are visible. Moreover, consistent with the linear theory, the RF Fourier power spectra of parasol (midget) cells, both in nonlinear simulations and experiments, specialize to cover low (high) spatial and high (low) temporal frequencies. Furthermore, we find that the parasol cells have a higher average firing rate than the midget cells (Fig. 4I), consistent with the greater sensitivity of parasol cells found both in biological data and our linear theory (see Fig. 3C). Also consistent with the linear theory, the neural network optimization loss is reduced for four cell types (two pairs) compared to two (one pair) (Fig. 4J). Moreover, the dependence of performance on cell type ratio mirrors the predictions of our linear theory (compare Fig 3DE with Fig. 4J and see Appendix H).

## 6 Discussion

In summary, we first demonstrated mathematically that there is a metabolic advantage to encoding natural movies with more than one convolutional cell type (Fig. 1C). By finding the optimal RFs,

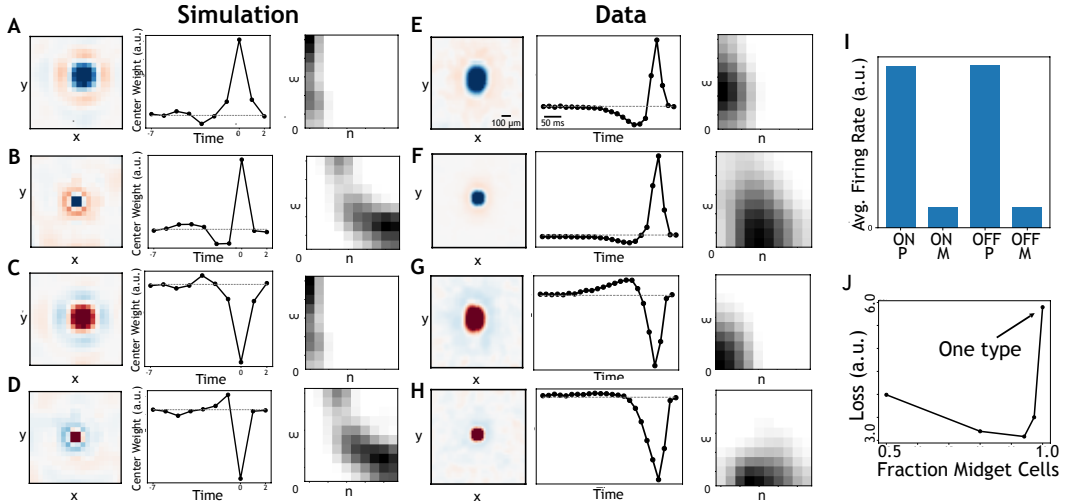

Figure 4: A nonlinear convolutional autoencoder reproduces primate retinal cell types. A, B, C, D) ON parasol-like type, ON midget-like type, OFF parasol-like type, and OFF midget-like type, respectively. Within each panel, far left: spatial receptive field (RF) at peak temporal slice of the spatiotemporal RF. Center: temporal RF, measured as the evolution of the central photoreceptor of the RF across time. Right: Space-time power spectrum of the RF, where dark shades correspond to high power. E, F, G, H) Same quantities measured from macaque retina (see Appendix I). I) Average firing rate per cell of each autoencoder cell type. J) Optimal loss (see Eqn. 3, p=1) as a function of the ratio between cell types densities. Note that optimal reconstruction with two cell type pairs is approximately 2 times better than with one cell type pair (black arrow).

strides and cell number ratios for the two populations, we show this advantage is substantial: a 33% reduction in RMS firing rate at a fixed encoding fidelity (reconstruction error: 0.5%). Moreover, the corresponding cell types have similar RFs and densities to midget and parasol cells. We also predict with great accuracy how the ratio of midget to parasol cells varies with the total cell density (Fig. 3F). Finally, by training a nonlinear neural network on the same task of reconstructing natural movies with a limited firing rate budget, we again confirm the advantage of having midget and parasol cells, and we find further differentiation into ON and OFF types.

There are a number of other ganglion cell types found in the primate retina [17] (20 types). Our current model accounts for the four most common cell types (ON and OFF midget and parasol cells), but it could be extended to account for more cell types. The next most common cell type found in the primate retina is the small bistratified type [5], which, unlike midget and parasol cells, pools from blue cones with an opposite polarity to red and green cones. Midget cells are color sensitive [40], a property that we do not account for in our current model, due to our focus on grayscale movies. By taking into account the spatiotemporal statistics of colors in natural movies, one can likely understand the division of labor between midget, parasol and small bistratified cells observed in primates.

Our theory predicts primate cell types well, but interestingly we could not find a good match in other species, such as mouse. The most numerous ganglion cell type in the mouse retina is a selective, non-linear feature detector (W3 cells [41]), thought to serve as an alarm system for overhead predators. Intriguingly, the retina may have evolved to detect behaviorally important predator cues in small animals [7] and efficiently and faithfully encode natural movies in larger animals. A recent study using a deep convolutional model of the visual system suggests that retinal computations either emerge as linear and information preserving encoders, or in the contrary as non-linear feature detectors, depending on the degree of neural resources allocated to downstream visual circuitry [42].

Thus overall our work suggests that the retina has evolved to efficiently encode the translation invariant statistics of natural movies through convolutional operations. Our model strikingly accounts for the 4 dominant cell types comprising 70% of all primate ganglion cells. Furthermore, promising extensions of this work to color statistics could expand the reach of this theory to encompass even greater cell-type diversity.

## Acknowledgements

We thank Alexandra Kling and E.J. Chichilnisky for useful discussions, and for providing us with receptive field visualizations of real midget and parasol cells. We thank Gabriel Mel for a helpful insight about the two-cell proof. We thank the Karel Urbanek Postdoctoral fellowship (S.O) and the NIH Brain Initiative U01-NS094288 (S.D), and the Burroughs-Wellcome, McKnight, James S. McDonnell, and Simons Foundations, and the Office of Naval Research (S.G) for support.

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
