[Supplementary Material]

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

## Footnotes

[3]We can show this using the identity of Eq. 40 where $m$ is replaced by $n_2$.

[4]Because the power decreases monotonically with frequency, the highest power image frequency corresponding to $m$ is simply given by $n_\text{M}(m) = m$.

[5]The optimum could have each ganglion cell spatial frequency pull from multiple photoreceptor spatial frequencies, where the spatial frequency pulled from was a function of the temporal frequency. This will not happen for natural movie statistics, since the power distribution monotonically decreases with both spatial and temporal frequency.

[6]This is because $\tilde{F}$ is the Fourier transform of the filter strength *scaled by the square root of the number of neurons*. The magnitude of the Fourier transform of $F_{\mathbb{A}}, F_{\mathbb{B}}$ have each increased by $\sqrt{2}$ for their respective modes. The intuition behind this is that when encoding a mode at fixed SNR with a variable number of neurons, the *total* power devoted to encoding that mode must remain constant. Therefore, when there are fewer neurons, each neuron must dedicate more power, and thus a stronger filter, to that mode.

[7] The number of modes selected will be equal to the number of cells of that type (see App. B2).

[8] This grid search assumes that in the optimum, there is no mode $n$ for which both $\tilde{F}_\mathbb{A}(n)$, $\tilde{F}_\mathbb{B}(n)$ are nonzero, i.e. no mode is shared between the two cell types. We find this to be empirically true from optimizing the linear problem numerically with gradient descent (code available online).

[9] We also use a variant which minimizes total firing rate without going below a minimal coding fidelity.

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

## Appendix

All code is available at https://github.com/ganguli-lab/RetinalCellTypes.

## Table Of Contents

# A Optimizing receptive fields for a dense convolutional array of ganglion cells

In this section, we present the mathematical framework that allows us to find the optimal receptive field for a convolutional array of ganglion cells encoding photoreceptor activations under a constraint on total firing rate [1]. We first solve the problem for one ganglion cell encoding the activation of one photoreceptor in App. A.1. We then use this result to solve for $N_{\mathcal{C}}$ ganglion cells of a single type $\mathcal{C}$ collectively encoding $N_p$ photoreceptor activations, where $N_{\mathcal{C}} = N_p$, in App. A.2.

## A.1 Linear reconstruction of a one dimensional signal from a single linear filter

Consider we have a scalar input X drawn from a Gaussian distribution with variance $\mathbf{C_{XX}}$:

$$\log(\mathbf{P}(X)) = -\frac{X^2}{\mathbf{C_{XX}}} + \text{const.} \tag{12}$$

This is measured using a neuron with a scalar filter F; this neuron also has intrinsic Gaussian noise $\eta$ with magnitude $\mathbf{C}_{\eta\eta} = \sigma_\eta^2$, yielding a firing rate of:

$$Y = F X + \eta \tag{13}$$

The reconstruction problem is as such: given priors of Eq. 12, which reconstructed signal $X_r$ is closest to the actual signal? Because all statistics are Gaussian, this corresponds to finding the most likely $X_r$ that produced the observed firing rate, which is given by Bayes rule:

$$\log(\mathbf{P}(X_r)|Y) + \text{const.} = \underbrace{-\frac{X_r^2}{2\mathbf{C_{XX}}}}_{\text{Priors}} \underbrace{-\frac{(FX_r - Y)^2}{2\mathbf{C}_{\eta\eta}}}_{\log(\mathbf{P}(\eta|X_r,Y))} = -\frac{X_r^2}{2}\left(\mathbf{C_{XX}^{-1}} + F^2\mathbf{C}_{\eta\eta}^{-1}\right) + \mathbf{C}_{\eta\eta}^{-1} F Y X_r \underbrace{-\frac{Y^2}{2\mathbf{C}_{\eta\eta}}}_{\text{Constant}} \tag{14}$$

We solve for $X_r$ by setting the partial derivative of log probability with respect to $X_r$ to be zero:

$$\frac{d\log(\mathbf{P}(X_r|Y))}{dX_r} = -\left(\mathbf{C_{XX}^{-1}} + F^2\mathbf{C}_{\eta\eta}^{-1}\right)X_r + \mathbf{C}_{\eta\eta}^{-1} Y F = 0 \tag{15}$$

Multiplying each side by $\mathbf{C}_{\eta\eta}\mathbf{C_{XX}}$, we get

$$-\left(\mathbf{C}_{\eta\eta} + F^2\mathbf{C_{XX}}\right)X_r + \mathbf{C_{XX}}YF = 0 \qquad \Rightarrow \qquad X_r = Y \cdot \frac{\mathbf{C_{XX}}F}{F^2\mathbf{C_{XX}} + \mathbf{C}_{\eta\eta}} \tag{16}$$

How much variance does the reconstructed signal explain? We note the reconstructed signal is uncorrelated with the reconstruction error, i.e.

$$\langle(X - X_r)X_r\rangle = 0 \tag{17}$$

Therefore, the variance explained is simply the variance of the reconstructed signal:

$$\mathcal{V} = \langle X^2\rangle - \langle(X - X_r)^2\rangle = 2\langle X X_r\rangle - \langle X_r^2\rangle = \langle X_r^2\rangle, \tag{18}$$

where we have used Eq. 17 in the last step. Using:

$$\langle Y^2\rangle = \langle[FX + \eta]^2\rangle = F^2\mathbf{C_{XX}} + \mathbf{C}_{\eta\eta}, \tag{19}$$

we calculate the variance of the reconstructed signal:

$$\mathcal{V} = \langle X_r^2\rangle = \langle Y^2\rangle \cdot \frac{F\mathbf{C_{XX}}^2 F}{(F\mathbf{C_{XX}}F + \mathbf{C}_{\eta\eta})^2} = \left(F^2\mathbf{C_{XX}} + \mathbf{C}_{\eta\eta}\right) \cdot \frac{F^2\mathbf{C_{XX}}^2}{\left(F^2\mathbf{C_{XX}} + \mathbf{C}_{\eta\eta}\right)^2} = \frac{F^2\mathbf{C_{XX}}^2}{\left(F^2\mathbf{C_{XX}} + \mathbf{C}_{\eta\eta}\right)} \tag{20}$$

**Optimal filter magnitude given a penalty on firing rate.**  Eq. 20 tells us the variance explained as a function of filter strength. Now consider optimizing the filter strength with a penalty term on the variance of firing rate. This corresponds to maximizing the following objective function:

$$\mathcal{O} = \mathcal{V} - \lambda\langle Y^2\rangle = \frac{F^2\mathbf{C_{XX}}^2}{\left(F^2\mathbf{C_{XX}} + \mathbf{C}_{\eta\eta}\right)} - \lambda\left(F^2\mathbf{C_{XX}} + \mathbf{C}_{\eta\eta}\right). \tag{21}$$

Defining $\mathcal{P}$ as the power spent encoding the input, $\mathcal{P} = \left(F^2\mathbf{C_{XX}}\right)$, this objective can be maximized as:

$$\frac{d\mathcal{O}}{d\mathcal{P}_{\text{Opt}}} = \mathbf{C_{XX}} \cdot \frac{d[\mathbf{C}_{\eta\eta}/\left(\mathcal{P}_{\text{Opt}} + \mathbf{C}_{\eta\eta}\right)]}{d\mathcal{P}_{\text{Opt}}} - \lambda = 0. \tag{22}$$

We simplify this as:

$$\frac{d\mathcal{O}}{d\mathcal{P}_{\text{Opt}}} = \mathbf{C}_{\text{XX}} \cdot \left[ \frac{\mathbf{C}_{\eta\eta}}{(\mathcal{P}_{\text{Opt}} + \mathbf{C}_{\eta\eta})^2} \right] - \lambda = 0 \tag{23}$$

$$\Rightarrow \mathbf{C}_{\text{XX}} \cdot \left[ \frac{\mathbf{C}_{\eta\eta}}{(\mathcal{P}_{\text{Opt}} + \mathbf{C}_{\eta\eta})^2} \right] = \lambda \tag{24}$$

$$\Rightarrow \mathbf{C}_{\text{XX}} \cdot \left[ \frac{\mathbf{C}_{\eta\eta}}{\lambda} \right] = (\mathcal{P}_{\text{Opt}} + \mathbf{C}_{\eta\eta})^2 \tag{25}$$

Because $\mathcal{P} \geq 0$, we arrive at our equation for the optimal power spent:

$$\Rightarrow \mathcal{P}_{\text{Opt}} = \left[ \sqrt{\frac{\mathbf{C}_{\text{XX}} \cdot \mathbf{C}_{\eta\eta}}{\lambda}} - \mathbf{C}_{\eta\eta} \right]_+ \tag{26}$$

where we use the notation:

$$[x]_+ = \left\{ \begin{array}{ll} x & x \geq 0 \\ 0 & x < 0. \end{array} \right. \tag{27}$$

Because $\mathbf{C}_{\text{XX}}, \mathbf{C}_{\eta\eta}$ are $1 \times 1$ matrices, we rewrite $\mathbf{C}_{\text{XX}} = \text{S}$, $\mathbf{C}_{\eta\eta} = \sigma_\eta^2$. Moreover, because $\text{F}^2 = \mathcal{P}/\text{S}$, we can build similar intuition for the filter magnitude:

$$(\text{F}_{\text{Opt}})^2 = \text{Q}^{-1} \left[ \mathcal{H} - \text{Q}^{-1} \right]_+, \text{ where } \text{Q} = \sqrt{\text{S}/\sigma_\eta^2}, \ \ \mathcal{H} = 1/\sqrt{\lambda} \tag{28}$$

where we call Q the "channel quality" to see that the volume of water (squared filter strength) devoted to encoding corresponds to the volume filled by filling a cylinder of area $\text{Q}^{-1}$ and height $\text{Q}^{-1}$ to a level of $\mathcal{H}$. The result of Eq. 28 will be used frequently, as the optimization of many convolutional neurons encoding many photoreceptors activations will reduce to it.

## A.2 Convolutional encoding where the number of neurons equals the number of photoreceptors

How do we best encode a photoreceptor image with a convolutional set of linear neurons? We are first going to assume that the number of neurons is equal to the number of photoreceptors, i.e. $\text{N}_\mathcal{C} = \text{N}_\text{p} = \text{N}$. In the next section, we then generalize to the case where there can be multiple photoreceptors for each neuron.

Each neuron, indexed by $j$, has a filter $\text{F}_j(i) = \text{F}(-i + j)$ and encodes a photoreceptor image of size N:

$$\text{Y}_j = \sum_{i=0}^{\text{N}-1} \text{F}(-i + j) \cdot \text{X}_i + \eta_j \tag{29}$$

where $\text{Y}_j$ is the neuron activation, $\text{X}_i$ is the activation of photoreceptor $i$, and $\eta_j$ is the noise at the output of neuron $j$.

It is beneficial to represent the photoreceptor activations, filters, and ganglion cell firing rates in Fourier space. We define the Fourier transform of photoreceptor activations as $\tilde{\text{X}}$, the Fourier transform of ganglion cell activations as $\tilde{\text{Y}}$, and the scaled Fourier transform of the filter as $\tilde{\text{F}}$:

$$\tilde{\text{X}}(n) = \frac{1}{\sqrt{\text{N}}} \sum_{i=0}^{\text{N}-1} \text{X}_i e^{(-2\pi \text{i})ni/\text{N}}, \ \ \tilde{\text{Y}}(m) = \frac{1}{\sqrt{\text{N}}} \sum_{j=0}^{\text{N}-1} \text{Y}_j e^{(-2\pi \text{i})mj/\text{N}}, \ \ \tilde{\text{F}}(n) = \sum_{i=0}^{\text{N}-1} \text{F}(i) e^{(-2\pi \text{i})ni/\text{N}}. \tag{30}$$

We define $\tilde{\text{F}}$ to be the Fourier transform of the filter scaled by $\sqrt{\text{N}}$ for convenience in future calculations.

Here we demonstrate the convolution theorem to show that the Fourier transform of the ganglion cell firing rate at a particular spatial frequency is simply the product of the photoreceptor activations and filters at that same frequency (Eq. 41).

The inverse Fourier transforms are:

$$\text{X}_i = \frac{1}{\sqrt{\text{N}}} \sum_{n=-\text{N}/2+1}^{\text{N}/2} \tilde{\text{X}}(n) e^{(2\pi \text{i})ni/\text{N}}, \ \ \text{Y}_j = \frac{1}{\sqrt{\text{N}}} \sum_{m=-\text{N}/2+1}^{\text{N}/2} \tilde{\text{Y}}(m) e^{(2\pi \text{i})mj/\text{N}}, \ \ \text{F}(i) = \frac{1}{\text{N}} \sum_{n=-\text{N}/2+1}^{\text{N}/2} \tilde{\text{F}}(n) e^{(2\pi \text{i})ni/\text{N}}. \tag{31}$$

We can write the signal component of the firing rate of neuron $j$ in terms of the Fourier modes of the photoreceptor activations and filter:

$$Y_j = \sum_i F(-i+j) \cdot X_i + \eta_j \qquad (32)$$

$$= \sum_i \left[ \frac{1}{N} \sum_{n=-N/2+1}^{N/2} \tilde{F}(n) e^{(2\pi i)n(-i)/N} e^{(2\pi i)nj/N} \right] \cdot \left[ \frac{1}{\sqrt{N}} \sum_{n_2=-N/2+1}^{N/2} \tilde{X}(n_2) e^{(2\pi i)n_2 i/N} \right] + \eta_j \qquad (33)$$

$$= N^{-3/2} e^{(2\pi i)nj/N} \sum_{n,n_2} \tilde{F}(n)\tilde{X}(n_2) \left[ \sum_i e^{(2\pi i)(n_2-n)i/N} \right] + \eta_j \qquad (34)$$

We may rewrite Eq. 34 as[3]:

$$Y_j = \frac{1}{\sqrt{N}} \sum_{n=-N/2+1}^{N/2} \tilde{F}(n) \cdot \tilde{X}(n) e^{(2\pi i)nj/N} + \eta_j \qquad (35)$$

Using this, the Fourier modes of firing rate $\tilde{Y}(m)$ (i.e. the Fourier transform applied on the convolutional array of neurons) can be written as:

$$\tilde{Y}(m) = \frac{1}{\sqrt{N}} \sum_{j=0}^{N-1} Y_j e^{(-2\pi i)mj/N} \qquad (36)$$

$$= \frac{1}{N} \sum_{j=0}^{N-1} \sum_{n=-N/2+1}^{N/2} \tilde{F}(n) \cdot \tilde{X}(n) e^{(2\pi i)nj/N} e^{(-2\pi i)mj/N} + \frac{1}{\sqrt{N}} \sum_j \eta_j e^{(-2\pi i)mj/N}. \qquad (37)$$

We define $\tilde{\eta}(m)$ as the Fourier transform of the noise:

$$\tilde{\eta}(m) = \frac{1}{\sqrt{N}} \sum_j \eta_j e^{(-2\pi i)mj/N_C}. \qquad (38)$$

We note that each $\tilde{\eta}(m)$ is an independent Gaussian also having variance $\sigma_\eta^2$ and can rearrange terms:

$$\tilde{Y}(m) = \frac{1}{N} \sum_n \tilde{F}(n) \cdot \tilde{X}(n) \sum_j e^{(2\pi i)(n-m)j/N_C} + \tilde{\eta}(m), \qquad (39)$$

and can use the identity:

$$\sum_{j=0}^{N-1} e^{(2\pi i)(n-m)j} = \begin{cases} N & n = m \\ 0 & \text{Otherwise} \end{cases} \qquad (40)$$

to see that the firing rate at a particular spatial frequency $\tilde{Y}_n$ is affected *only* by the photoreceptor image at that same frequency:

$$\tilde{Y}(m) = \delta_{n,m} \tilde{F}(n) \cdot \tilde{X}(n) + \tilde{\eta}(m), \qquad (41)$$

where $\delta_{n,m}$ is the Kronecker delta function.

To find the optimal filter strengths, we note that when we add a $\lambda$ penalty to the total squared firing rate $\sum_j \langle Y_j^2 \rangle$, the objective function becomes:

$$\mathcal{O} = \sum_n \left( \frac{|\tilde{F}(n)|^2 S(n)^2}{|\tilde{F}(n)|^2 S(n) + \sigma_\eta^2} - \lambda \cdot \left( S(n)|\tilde{F}(n)|^2 + \sigma_\eta^2 \right) \right) \qquad (42)$$

and we see that the objective function Eq. 42 consists of many non-interacting components, each corresponding to a particular spatial frequency $n$, summed together. Moreover, each component corresponding to ganglion cell spatial frequency $n$ is equivalent to the objective maximized in Eq. 21. Therefore, the optimal filters for Eq. 42 are given by Eq. 28:

$$|\tilde{F}_{\text{Opt}}(n)|^2 = Q^{-1}(n) \left[ \mathcal{H} - Q^{-1}(n) \right]_+, \quad \text{where } Q(n) = \sqrt{S(n)/\sigma_\eta^2}, \ \mathcal{H} = 1/\sqrt{\lambda}. \qquad (43)$$

# B Optimal receptive fields for arbitrary ganglion cell density

In App. A, we found the optimal receptive fields for a convolutional array of ganglion cells where the number of ganglion cells was equal to the number of photoreceptors [1]. Now we will solve the more realistic case where the number of ganglion cells is less than the number of photoreceptors. For this, we will first solve the simplified case of one neuron encoding a multidimensional signal (App. B.1), to show that the best single-neuron filter should align with the strongest eigenmode of the covariance. Then, using this result for convolutional encoding with fewer ganglion cells than photoreceptors, we will show that the optimal convolutional filter selects only the strongest $N_C$ modes (thus avoiding aliasing [33]), where $N_C$ is the number of ganglion cells. The filter strengths are analogous to those solved in App. A.2.

Figure 5: Road map of App. B. A1, A2) When a single neuron can encode multiple inputs (Eq. 45), the optimal filter will be aligned with the largest eigenmode of the input covariance (Eq. 54). B) In our framework, the number of neurons can be smaller than the number of photoreceptors. Therefore, multiple Fourier modes $\tilde{X}(n)$ will map onto the same firing modes $\tilde{Y}(m)$. C) Representation of this mapping (Eq. 63). While ganglion cell spatial modes (lower dots) pull from many photoreceptor modes (upper dots), each photoreceptor mode maps to only a *single* ganglion cell spatial mode. D) Therefore, we may reduce this problem to many many non-interacting instances of (A) to show the optimal convolutional filter (Eq. 66) will not be aliased, and each ganglion cell spatial frequency will pull from a single photoreceptor spatial frequency.

## B.1 Linear reconstruction of a multidimensional signal from a single linear filter

This section is a generalization of App. A.1 to a single neuron encoding a multidimensional input.

Consider we have a *vector* input $\vec{X}$ drawn from a Gaussian with covariance $\mathbf{C}_{XX}$:

$$\log(\mathbf{P}(\vec{X})) = -\frac{1}{2}\vec{X}^T \mathbf{C}_{XX}^{-1}\vec{X} + \text{const.} \tag{44}$$

We measure this input using a neuron with receptive field $\vec{F}$ to have a firing rate of:

$$Y = \vec{X}^T \cdot \vec{F} + \eta. \tag{45}$$

The reconstruction problem is as such: given priors of Eq. 44, solve which reconstructed signal $\vec{X}_r$ is closest to the actual signal. Because all statistics are Gaussian, this corresponds to finding the most likely $\vec{X}_r$ that produced the observed firing rate:

$$\log(\mathbf{P}(\vec{X}_r)|Y) + \text{const.} = \underbrace{-\frac{1}{2}\vec{X}_r^T \mathbf{C}_{XX}^{-1}\vec{X}_r}_{\log(\mathbf{P}(\vec{X}_r))} - \underbrace{\frac{1}{2}\mathbf{C}_{\eta\eta}^{-1}\left(\vec{F}^T\vec{X}_r - Y\right)^2}_{\log(\mathbf{P}(\eta|\vec{X}_r, Y))} \tag{46}$$

$$= -\frac{1}{2}\vec{X}_r^T\left[\mathbf{C}_{XX}^{-1} + \vec{F}\mathbf{C}_{\eta\eta}^{-1}\vec{F}^T\right]\vec{X}_r + \mathbf{C}_{\eta\eta}^{-1}Y \times \vec{F}^T\vec{X}_r - \underbrace{\frac{1}{2}\mathbf{C}_{\eta\eta}^{-1}Y^2}_{\text{Constant}} \tag{47}$$

We solve for $\vec{X}_r$ by setting the gradient of log probability with respect to $\vec{X}_r$ to be zero:

$$\frac{d\log\left(\mathbf{P}(\vec{X}_r|Y)\right)}{d\vec{X}_r} = -\left(\mathbf{C}_{XX}^{-1} + \vec{F}\mathbf{C}_{\eta\eta}^{-1}\vec{F}^T\right)\vec{X}_r + \mathbf{C}_{\eta\eta}^{-1}Y\vec{F} = 0 \quad \Rightarrow \quad \vec{X}_r = Y \cdot \frac{\mathbf{C}_{XX}\vec{F}}{\vec{F}^T\mathbf{C}_{XX}\vec{F} + \mathbf{C}_{\eta\eta}} \tag{48}$$

Analogously to App. 18, the magnitude of variance explained is simply the variance of the guess vector $\vec{X}_r$. Using:

$$\langle Y^2 \rangle = \left\langle \left[\vec{F}^T\vec{X} + \eta\right]^2 \right\rangle = \vec{F}^T\mathbf{C}_{XX}\vec{F} + \mathbf{C}_{\eta\eta} \tag{49}$$

we calculate the variance explained:

$$\mathcal{V} = \left\langle |\vec{X_r}|^2 \right\rangle = \left( \vec{F}^T \mathbf{C}_{XX} \vec{F} + \mathbf{C}_{\eta\eta} \right) \cdot \left\langle Y^2 \right\rangle \cdot \frac{\vec{F}^T \mathbf{C}_{XX}^2 \vec{F}}{\left[ \vec{F}^T \mathbf{C}_{XX} \vec{F} + \mathbf{C}_{\eta\eta} \right]^2} = \left[ \vec{F}^T \mathbf{C}_{XX} \vec{F} + \mathbf{C}_{\eta\eta} \right] \cdot \frac{\vec{F}^T \mathbf{C}_{XX}^2 \vec{F}}{\left[ \vec{F}^T \mathbf{C}_{XX} \vec{F} + \mathbf{C}_{\eta\eta} \right]^2} = \frac{\vec{F}^T \mathbf{C}_{XX}^2 \vec{F}}{\left[ \vec{F}^T \mathbf{C}_{XX} \vec{F} + \mathbf{C}_{\eta\eta} \right]} \quad (50)$$

What is the optimal multidimensional filter $\vec{F}$ that maximizes variance explained under a penalty on firing rate? Below, we show that this multidimensional problem reduces to a one-dimensional problem by proving that the optimal $\vec{F}$ must be a multiple of the largest principal component of $\mathbf{C}_{XX}$. Rewriting the $1 \times 1$ matrix $\mathbf{C}_{\eta\eta}$ as $\sigma_\eta^2$, in the eigenspace of $\mathbf{C}_{XX}$, Eq. 50 can be written as:

$$\mathcal{V} = \frac{\sum_n F(n)^2 S(n)^2}{\sigma_\eta^2 + \sum_n F(n)^2 S(n)}, \quad (51)$$

where the eigenvectors of $\mathbf{C}_{XX}$ are indexed by $n$, and have eigenvalues $S(n)$. Calling the mode index with highest power $n_0$, we may write the objective as:

$$\mathcal{O} = \mathcal{V} - \lambda \left\langle Y^2 \right\rangle = \frac{\sum_n \left( F(n)^2 S(n) \right) S(n_0) + \overbrace{\sum_n F(n)^2 S(n) \left( S(n) - S(n_0) \right)}^{\leq 0}}{\sigma_\eta^2 + \sum_n F(n)^2 S(n)} - \lambda \left( \sigma_\eta^2 + \sum_n F(n)^2 S(n) \right) \quad (52)$$

The right term of the numerator can only hurt performance; if there is some other frequency $n \neq n_0$ with nonzero filters, we note that the transformation redistributing power from mode $n$ to $n_0$:

$$F(n)^2 \to F(n)^2 - \delta/S(n), \qquad F(n_0)^2 \to F(n_0)^2 + \delta/S(n_0) \quad (53)$$

increases the right term of numerator by $\delta \cdot (S(n_0) - S_n)$ without affecting the total power $\sum F(n)^2 S(n)$, and thus will not affect the rest of the equation. Therefore, the projection should be mapped to only the *largest* mode of the covariance matrix (Fig. 5A), saturating at a performance of:

$$\mathcal{O} = \frac{F(n_0)^2 S(n_0)^2}{(F(n_0)^2 S(n_0)) + \sigma_\eta^2} - \lambda \left( F^2(n_0) S(n_0) + \sigma_\eta^2 \right) \quad (54)$$

which is optimized in the same manner as the scalar case of App. A.1 to yield

$$\left( F_{\text{Opt}} \right)(n)^2 = \delta_{n,n_0} \, Q^{-1} \left[ \mathcal{H} - Q^{-1} \right]_+, \text{ where } Q = \sqrt{S(n_0)/\sigma_\eta^2}, \quad \mathcal{H} = 1/\sqrt{\lambda} \quad (55)$$

i.e., the 1D optimum after projecting onto the largest eigenmode $n_0$.

## B.2 Optimal convolutional encoding with fewer ganglion cells than photoreceptors

When there are fewer ganglion cells than photoreceptors, the above framework of App. A.2 must only be slightly modified. Now, instead of one ganglion cell per photoreceptor, we assume a convolutional stride $s$ greater than one, so that each ganglion cell is $s$ apart from its nearest neighbor, yielding a total of $N_p/s$ ganglion cells (we assume $N_p$ is a multiple of $s$). Consider a ganglion cell $j$, centered about $js$, with filter $F_j(i) = F(-i + sj)$. The firing rate of neuron $j$ will be:

$$Y_j = \sum_i F(-i + sj) \cdot X_i + \eta_j, \quad (56)$$

where $Y_j$ is the neuron activation, $X_i$ is the activation of photoreceptor $i$, and $\eta_j$ is the noise at the output of neuron $j$.

It is beneficial to represent the photoreceptor activations, filters, and ganglion cell firing rates in Fourier space. We define the Fourier transform of photoreceptor activations as $\tilde{X}$, the Fourier transform of ganglion cell activations as $\tilde{Y}$, and the scaled Fourier transform of the filter as $\tilde{F}$:

$$\tilde{X}(n) = \frac{1}{\sqrt{N_p}} \sum_{i=0}^{N_p - 1} X_i e^{(-2\pi i)ni/N_p}, \quad \tilde{Y}(m) = \frac{1}{\sqrt{N_C}} \sum_{j=0}^{N_C - 1} Y_j e^{(-2\pi i)mj/N_C}, \quad \tilde{F}(n) = \sqrt{N_C} \times \frac{1}{\sqrt{N_p}} \sum_{i=0}^{N_p - 1} F(i) e^{(-2\pi i)ni/N_p}$$

We define $\tilde{F}$ to be the Fourier transform of the filter scaled by $\sqrt{N_C}$ for convenience in future calculations.

We can use the convolution theorem to write the signal component of the firing rate of neuron $j$ in terms of the Fourier modes of the photoreceptor activations and filter:

$$\text{Y}_j = \sum_i \text{F}(-i + sj) \cdot \text{X}_i + \eta_j \qquad = \frac{1}{\sqrt{\text{N}_\mathcal{C}}} \sum_{n=-\text{N}_\text{p}/2+1}^{\text{N}_\text{p}/2} \tilde{\text{F}}(n) \cdot \tilde{\text{X}}(n) e^{(2\pi\text{i})njs/\text{N}_\text{p}} + \eta_j \qquad (57)$$

The Fourier modes of firing rate $\tilde{\text{Y}}(m)$ (i.e. the Fourier transform applied on the convolutional array of ganglion cells) can be translated to photoreceptor space:

$$\tilde{\text{Y}}(m) = \frac{1}{\sqrt{\text{N}_\mathcal{C}}} \sum_{j=0}^{\text{N}_\mathcal{C}-1} \text{Y}_j e^{(-2\pi\text{i})msj/\text{N}_\text{p}} \qquad (58)$$

where $\text{N}_\mathcal{C} = \text{N}_\text{p}/s$, i.e. the total number of neurons is equal to the number of photoreceptors times the neuron-to-photoreceptor ratio. We rewrite the Fourier transform of firing rates in terms of the Fourier transform of photoreceptor activations and convolutional filter:

$$\tilde{\text{Y}}(m) = \frac{1}{\text{N}_\mathcal{C}} \sum_j \sum_n \tilde{\text{F}}(n) \cdot \tilde{\text{X}}(n) e^{(2\pi\text{i})nsj/\text{N}_\text{p}} e^{(-2\pi\text{i})msj/\text{N}_\text{p}} + \frac{1}{\sqrt{\text{N}_\mathcal{C}}} \sum_j \eta_j e^{(-2\pi\text{i})mj/\text{N}_\mathcal{C}} \qquad (59)$$

We define $\tilde{\eta}(m)$ as the Fourier transform of the noise:

$$\tilde{\eta}(m) = \frac{1}{\sqrt{\text{N}_\mathcal{C}}} \sum_j \eta_j e^{(-2\pi\text{i})mj/\text{N}_\mathcal{C}} \qquad (60)$$

and note that each $\tilde{\eta}(m)$ is an independent Gaussian also having variance $\sigma_\eta^2$. We can rearrange terms:

$$\tilde{\text{Y}}(m) = \frac{1}{\text{N}_\mathcal{C}} \sum_n \tilde{\text{F}}(n) \cdot \tilde{\text{X}}(n) \sum_j e^{(2\pi\text{i})(n-m)sj/\text{N}_\text{p}} + \tilde{\eta}(m) \qquad (61)$$

Then, we generalize the identity of Eq. 40 to:

$$\sum_{j=0}^{\text{N}_\mathcal{C}-1} e^{(2\pi\text{i})(n-m)sj/\text{N}_\text{p}} = \begin{cases} \text{N}_\mathcal{C} & m - n \mod \text{N}_\mathcal{C} = 0 \\ 0 & \text{Otherwise} \end{cases} \qquad (62)$$

and see that $\tilde{\text{Y}}(m)$ is affected by all $\tilde{\text{X}}(n)$ where $[m-n] \mod \text{N}_\mathcal{C} = 0$. This yields the most simplified (Fig. 5C) form of the firing rate in Fourier space:

$$\tilde{\text{Y}}(m) = \sum_{n=m+\text{n'}\text{N}_\mathcal{C}} \tilde{\text{F}}(n) \cdot \tilde{\text{X}}(n) + \tilde{\eta}(m). \qquad (63)$$

**Optimal encoding.** The objective (Eq. 3) given by this encoding for $\text{p} = 2$ is:

$$\mathcal{O} = \sum_{m=-\text{N}_\mathcal{C}/2+1}^{\text{N}_\mathcal{C}/2} \left( \frac{\sum_{n=m+\text{n'}\text{N}_\mathcal{C}} \left|\tilde{\text{F}}(n)\right|^2 \cdot \text{S}(n)^2}{\sigma_\eta^2 + \sum_{n=m+\text{n'}\text{N}_\mathcal{C}} \left|\tilde{\text{F}}(n)\right|^2 \text{S}(n)} \right) - \lambda \left( \text{N}_\mathcal{C}\sigma_\eta^2 + \sum_{n=0}^{\text{N}_\text{p}-1} \left( \left|\tilde{\text{F}}(n)\right|^2 \cdot \text{S}(n) \right) \right) \qquad (64)$$

We rewrite Eq. 64 as:

$$\mathcal{O} = \sum_m \left( \frac{\sum_{n=m+\text{n'}\text{N}_\mathcal{C}} \left|\tilde{\text{F}}(n)\right|^2 \text{S}(n)^2}{\sigma_\eta^2 + \sum_{n=m+\text{n'}\text{N}_\mathcal{C}} \left|\tilde{\text{F}}(n)\right|^2 \text{S}(n)} - \lambda \cdot \left[ \sigma_\eta^2 + \sum_{n=m+\text{n'}\text{N}_\mathcal{C}} \text{S}(n) \left|\tilde{\text{F}}(n)\right|^2 \right] \right) \qquad (65)$$

We can verify this step by inspection by noting that each $n$ penalty term is summed over exactly once in both Eq. 64 and Eq. 65.

Examining Eq. 65, we see that the objective function consists of many non-interacting components, each corresponding to a particular ganglion cell spatial frequency $m$, summed together. Moreover, each component corresponding to ganglion cell spatial frequency $m$ is equivalent to the objective maximized in Eq. 51. Because, as we saw in App. B.1, it is optimal for a single filter to draw *only* from the largest principal component, the optimal convolutional filter has $\tilde{\text{F}}(n) = 0$ for all $n \neq m$ (Fig. 5D)[4]. Therefore, the optimal filter in frequency

space is simply a scaled, truncated version of Eq. 43:

$$|\tilde{F}_{\text{Opt}}(n)|^2 = H\left(N_{\mathcal{C}}/2 - |n|\right) Q^{-1}(n)\left[\mathcal{H} - Q^{-1}(n)\right]_+, \text{ where } Q(n) = \sqrt{S(n)/\sigma_\eta^2}, \ \mathcal{H} = \frac{1}{\sqrt{\lambda}}, \quad (66)$$

where H is the Heaviside function, which ensures every ganglion cell spatial frequency draws only from a single photoreceptor spatial frequency.

## C   Generalization to encoding of natural movies with spatiotemporal filters

We derived the optimal *spatial* receptive fields for encoding natural *images* in App. B.2. How can we generalize this result to optimal *spatiotemporal* receptive fields for encoding natural *movies*? Now the firing rate of ganglion cell $j$ at particular discrete time $t$ will be given by:

$$Y_j(t) = \sum_{i=0}^{N_p} \sum_{t=0}^{T} F_j(i, t - t') \cdot X_i(t') + \eta_j(t) = \sum_{i=0}^{N_p} \sum_{t=0}^{T} F(-i + js, t - t') \cdot X_i(t') + \eta_j(t) \quad (67)$$

Where T is the length of the movie in frames, and for mathematical convenience we define both the receptive fields and the natural movie to be periodic in space and time.

It is beneficial to represent the photoreceptor activations, filters, and ganglion cell firing rates in Fourier space *and* time. We define the Fourier transform of photoreceptor activations as $\tilde{X}$, the Fourier transform of ganglion cell activations as $\tilde{Y}$, and the scaled Fourier transform of the filter as $\tilde{F}$:

$$\tilde{X}(n, \omega) = \frac{1}{\sqrt{N_p\, T}} \sum_{i=0}^{N_p-1} \sum_{t=0}^{T-1} X_i(t) e^{(-2\pi i)ni/N_p} e^{(-2\pi i)\omega t/T}, \quad (68)$$

$$\tilde{Y}(m, \omega) = \frac{1}{\sqrt{N_{\mathcal{C}}\, T}} \sum_{j=0}^{N_{\mathcal{C}}-1} \sum_{t=0}^{T-1} Y_j(t) e^{(-2\pi i)msj/N_p} e^{(-2\pi i)\omega t/T}, \quad (69)$$

$$\tilde{F}(n, \omega) = \sqrt{N_{\mathcal{C}} \times T} \times \frac{1}{\sqrt{N_p\, T}} \sum_{i=0}^{N_p-1} \sum_{t=0}^{T-1} F(i, t) e^{(-2\pi i)ni/N_p} e^{(-2\pi i)\omega t/T}, \quad (70)$$

where $t$ is an integer representing the frame of the movie, and $\omega$ is an integer representing the mode number in time. Each $\omega$ translates into a temporal frequency $2\pi\omega/T$. We define $\tilde{F}$ to be the Fourier transform of the filter scaled by $\sqrt{N_{\mathcal{C}} \times T}$ for convenience in future calculations.

This yields the encoding equation in Fourier space:

$$\tilde{Y}(m, \omega) = \sum_{n = m + \text{n'}N_{\mathcal{C}}} \tilde{F}(n, \omega) \cdot \tilde{X}(n, \omega) + \tilde{\eta}(m, \omega) \quad (71)$$

where $\tilde{\eta}(m, \omega)$ is gaussian with variance $\sigma_\eta^2$. The objective function becomes:

$$\mathcal{O} = \sum_m \sum_\omega \left( \frac{\sum_{n=m+\text{n'}N_{\mathcal{C}}} |\tilde{F}(n,\omega)|^2 S(n,\omega)^2}{\sigma_\eta^2 + \sum_{n=m+\text{n'}N_{\mathcal{C}}} |\tilde{F}(n,\omega)|^2 S(n,\omega)} - \lambda \cdot \left[ \sigma_\eta^2 + \sum_{n=m+\text{n'}N_{\mathcal{C}}} S(n,\omega) |\tilde{F}(n,\omega)|^2 \right] \right) \quad (72)$$

This is nearly identical to the previous case (Eq. 63), except that we have added a time mode. This may be optimized in the same manner[5] to yield:

$$|\tilde{F}_{\text{Opt}}(n,\omega)|^2 = H\left(N_{\mathcal{C}}/2 - |n|\right) Q^{-1}(n,\omega)\left[\mathcal{H} - Q^{-1}(n,\omega)\right]_+, \text{ where } Q(n,\omega) = \sqrt{S(n,\omega)/\sigma_\eta^2}, \ \mathcal{H} = \frac{1}{\sqrt{\lambda}}. \ (73)$$

# D  General proof of the benefit of multiple cell types

We will now show that two convolutional types can replicate the encoding fidelity of one convolutional type with less firing rate. For mathematical convenience, we assume an even number of photoreceptors. Given a single convolutional cell type $\mathbb{1}$ with optimal filters $\tilde{F}(n)_{\mathbb{1}}$, the RMS firing rate (i.e. $p = 1$ for Eq. 3) summed over all ganglion cells and all frames in the movie is:

$$\sum_{j=0}^{N_C^{\mathbb{1}}-1} \sum_{t} \sqrt{\langle Y_{j,\mathbb{1}}(t)^2 \rangle} = \sqrt{N_C^{\mathbb{1}} \times T} \sqrt{\sum_{j,t} \langle Y_{j,\mathbb{1}}(t)^2 \rangle} = \tag{74}$$

$$\sqrt{N_C^{\mathbb{1}} \times T} \sqrt{\sum_{m,\omega} \langle |\tilde{Y}_{\mathbb{1}}(n,\omega)|^2 \rangle} = \sqrt{N_C^{\mathbb{1}} \times T} \times \sqrt{\sum_{n,\omega} \left( \left| \tilde{F}_{\mathbb{1}}(n,\omega) \right|^2 \cdot S(n,\omega) + \sigma_\eta^2 \right)} \tag{75}$$

In the main paper and App. E, this constant and the movie length T are factored into the penalty term $\lambda$ and thus do not appear in the equations.

We can replace one convolutional type with two convolutional types; the first type, $\mathbb{A}$ covers $\mathcal{N}_{\mathbb{A}}$, the first half of all spatial frequencies ($|n| \leq N_C^{\mathbb{1}}/4$), the second type $\mathbb{B}$ covers $\mathcal{N}_{\mathbb{B}}$, the second half of all spatial frequencies ($N_C^{\mathbb{1}}/4 < |n| \leq N_C^{\mathbb{1}}/2$). The total number of ganglion cells remains the same, and the stride is doubled: $N_C^{\mathbb{A}} = N_C^{\mathbb{B}} = N_C^{\mathbb{1}}/2$, $s^{\mathbb{A}} = s^{\mathbb{B}} = 2s^{\mathbb{1}}$. The filters are changed to divide the frequencies between the two cell types.

$$\tilde{F}_{\mathbb{A}}(n,\omega) = \begin{cases} \tilde{F}_{\mathbb{1}}(n,\omega) & n \in \mathcal{N}_{\mathbb{A}} \\ 0 & n \in \mathcal{N}_{\mathbb{B}} \end{cases} \quad , \qquad\qquad \tilde{F}_{\mathbb{B}}(n,\omega) = \begin{cases} 0 & n \in \mathcal{N}_{\mathbb{A}} \\ \tilde{F}_{\mathbb{1}}(n,\omega) & n \in \mathcal{N}_{\mathbb{B}} \end{cases}$$

Note that although $\tilde{F}_{\mathbb{A}}, \tilde{F}_{\mathbb{B}}$ each equal $\tilde{F}_{\mathbb{1}}$, they each have a stronger filter within their respective modes [6].

The two cell types presented above have equal variance explained to the single type, as can be seen from the left half of Eq. 72.

When we calculate the total RMS rate of the first cell type:

$$\sum_{j=0}^{N_C^{\mathbb{A}}-1} \sum_{t=0}^{T-1} \langle |Y_{j,\mathbb{A}}(t)| \rangle = \sqrt{N_C^{\mathbb{1}} \times T} \times \frac{1}{\sqrt{2}} \times \sqrt{\sum_{n \in \mathcal{N}_{\mathbb{A}}} \sum_{\omega} \left( |\tilde{F}_{\mathbb{A}}(n,\omega)|^2 \cdot S(n,\omega) + \sigma_\eta^2 \right)} \tag{76}$$

and for the second cell type

$$\sum_{j=0}^{N_C^{\mathbb{B}}-1} \sum_{t=0}^{T-1} \langle |Y_{j,\mathbb{B}}(t)| \rangle = \sqrt{N_C^{\mathbb{1}} \times T} \times \frac{1}{\sqrt{2}} \times \sqrt{\sum_{n \in \mathcal{N}_{\mathbb{B}}} \sum_{\omega} \left( |\tilde{F}_{\mathbb{B}}(n,\omega)|^2 \cdot S(n,\omega) + \sigma_\eta^2 \right)} \tag{77}$$

and rewrite the total RMS firing rate for one convolutional cell type as:

$$\sum_{j=0}^{N_C^{\mathbb{1}}-1} \sum_{t=0}^{T} \langle |Y_{j,\mathbb{1}}(t)| \rangle = \sqrt{N_C^{\mathbb{1}} \times T} \times \sqrt{ \begin{array}{c} \sum_{n \in \mathcal{N}_{\mathbb{A}}} \sum_{\omega} \left( |\tilde{F}_{\mathbb{1}}(n,\omega)|^2 \cdot S(n,\omega) + \sigma_\eta^2 \right) + \\ \sum_{n \in \mathcal{N}_{\mathbb{B}}} \sum_{\omega} \left( |\tilde{F}_{\mathbb{1}}(n,\omega)|^2 \cdot S(n,\omega) + \sigma_\eta^2 \right) \end{array} } \tag{78}$$

we notice that right square root term for Eq. 78 is simply the sum of the right square root terms for Eq. 76 and Eq. 77. Because the square root of the mean is larger than the mean of the square root, i.e. $\sqrt{(A + B)/2} > (\sqrt{A} + \sqrt{B})/2$ when $A \neq B$ (when power monotonically decreases with frequency, $A > B$), we can show that two convolutional types can achieve *equal* reconstruction error with *less* RMS firing rate. Conversely, they can achieve *better* reconstruction error with the *same* total RMS firing rate.

It is not essential that the mean absolute value of firing rate was penalized. When the penalty on firing rate scales as a power smaller than two, multiple cell types will achieve the same encoding with a smaller penalty, i.e.

$$\sum_{j=0}^{N_C^\mathbb{A}-1} \sum_t \left\langle Y_{j,\mathbb{A}}(t)^2 \right\rangle^{p/2} + \sum_{j=0}^{N_C^\mathbb{B}-1} \sum_t \left\langle Y_{j,\mathbb{B}}(t)^2 \right\rangle^{p/2} < \sum_{j=0}^{N_C^\mathbb{1}-1} \sum_t \left\langle Y_{j,\mathbb{1}}(t)^2 \right\rangle^{p/2}$$

so long as $p < 2$. We note that this proof construction can be extended to show improvement for arbitrary ratios between the two cell types, as long as $(N_C)_\mathbb{A} + (N_C)_\mathbb{B} = (N_C)_\mathbb{1}$.

**Many Cell Types**

Extending this logic to equally dividing up frequency space among many cell types, the total firing rate becomes:

$$\sqrt{\frac{N_C^\mathbb{1} \times T}{N_{\text{Types}}}} \times \sum_\mathcal{C} \sqrt{\sum_{n \in \mathcal{N}_\mathcal{C}} \sum_\omega \left( |\tilde{F}_\mathcal{C}(n,\omega)|^2 \cdot S(n,\omega) + \sigma_\eta^2 \right)} \tag{79}$$

There the total firing of one type likewise can be rewritten as:

$$\sqrt{N_C^\mathbb{1} \times T} \times \sqrt{\sum_\mathcal{C} \left[ \sum_{n \in \mathcal{N}_\mathcal{C}} \sum_\omega \left( |\tilde{F}_\mathcal{C}(n,\omega)|^2 \cdot S(n,\omega) + \sigma_\eta^2 \right) \right]} \tag{80}$$

where $\sum_{n \in \mathcal{N}_\mathcal{C}}$ corresponds to the range of spatial frequencies covered by each cell type. We notice that the right square root term in Eq. 80 is simply the sum of the individual right square root terms in Eq. 79. Again, because the square root of the mean is larger than the mean of the square root, many convolutional types achieve the same encoding with a smaller total firing rate.

Because many convolutional types can achieve *equal* reconstruction error with *fewer* total RMS firing rate, they can achieve *better* reconstruction error with the *same* firing rate. In practice, there are diminishing returns with the number of unique cell types.

# E   Objective function and optimization procedure for two cell types

For two cell types, we assume that cells of different types don't share spatiotemporal modes, yielding an objective function of:

$$\mathcal{O} = \mathcal{V}^A - \lambda N_C^\mathbb{A} \sqrt{\langle Y_\mathbb{A}^2 \rangle} + \mathcal{V}^B - \lambda N_C^\mathbb{B} \sqrt{\langle Y_\mathbb{B}^2 \rangle} \tag{81}$$

We may replace identically weighted $\ell_1$ norms on the firing rates of type A and type B cells with differently weighted $\ell_2$ norms.

$$\mathcal{O} = \mathcal{V}^A - \lambda^\mathbb{A} N_C^\mathbb{A} \langle Y_\mathbb{A}^2 \rangle + \mathcal{V}^B - \lambda^\mathbb{B} N_C^\mathbb{B} \langle Y_\mathbb{B}^2 \rangle \tag{82}$$

While we know which filters at *fixed strides* $s^\mathbb{A}$, $s^\mathbb{B}$ and *fixed $\ell_2$ penalties* $\lambda^\mathbb{A}$, $\lambda^\mathbb{B}$ optimize Eq. 82, we do not know which combination of parameters optimizes these at a *fixed firing budget* (called $FR_{\text{Max}}$). To do so, we perform a grid search.

For each combination of $s^\mathbb{A}$, $s^\mathbb{B}$, $\lambda^\mathbb{A}$, $\lambda^\mathbb{B}$, we optimize the RFs for each type separately. We assign to type A the $N_p/s^\mathbb{A}$ lower spatial modes and to type B the $N_p/s^\mathbb{B}$ next modes, as we know that the best encoding scheme will (1) pool only from the modes with highest power and will limit the number of modes selected [7] to avoid aliasing, (2) will avoid mixing modes between cell types [8], and (3) will favor a solution with a maximal asymmetry between the power of the modes associated to each cell type, so as to maximise the benefit of having two cell types.

We then compute the corresponding reconstruction error and total firing rate. The optimal scheme is the one that minimizes reconstruction error but does not exceed a total firing rate budget [9] that we define in advance. We fasten the optimization procedure by working in log space for spatial and temporal frequencies. The code is available online at https://github.com/ganguli-lab/RetinalCellTypes.

Below is a pseudo-code version of the algorithm to maximize variance explained under a fixed RMS firing rate budget of $\mathrm{FR_{Max}}$:

```
 1: function GRIDSEARCH(FR_Max, S(n, ω)))
 2:     V_Best ← 0
 3:     for all s^𝔸, s^𝔹, λ^𝔸, λ^𝔹 do
 4:         Resets the calculated filters, variance explained, and power usage
 5:         𝒫^𝔸_Tot ← 0
 6:         𝒱^A ← 0
 7:         𝒫^𝔹_Tot ← 0
 8:         𝒱^B ← 0
 9:         for all n, ω do
10:             F̃_𝔸(n, ω) ← 0
11:             F̃_𝔹(n, ω) ← 0
12:         end for
13:
14:         Solve for the power usage and variance explained for cell type A
15:         for all n ∈ 𝒩_𝔸, all ω do
16:             F̃_𝔸(n, ω) ← F̃_Opt (S(n, ω), λ^𝔸)
17:             𝒫^𝔸_Tot ← 𝒫^𝔸_Tot + |F̃_𝔸(n, ω)|²S(n, ω) + σ²_η
18:             𝒱^A ← 𝒱^A + 𝒱 (F̃_𝔸(n, ω)S(n, ω), σ²_η)
19:         end for
20:
21:         Solve for the power usage and variance explained for cell type B
22:         for all n ∈ 𝒩_𝔹, all ω do
23:             F̃_𝔹(n, ω) ← F̃_Opt (S(n, ω), λ^𝔹)
24:             𝒫^𝔹_Tot ← 𝒫^𝔹_Tot + |F̃_𝔹(n, ω)|²S(n, ω)
25:             𝒱^B ← 𝒱^B + 𝒱 (F̃_𝔹(n, ω)S(n, ω), σ²_η)
26:         end for
27:
28:         if √(𝒫^𝔸_Tot N^𝔸_C) + √(𝒫^𝔹_Tot N^𝔹_C) ≤ FR_Max then
29:             if 𝒱^A + 𝒱^B > 𝒱_Best then
30:                 If the stride and penalties fall under the RMS firing rate budget and explain more variance
31:                 than any other solution, they become our newest candidate solution. Note that the total
32:                 firing rate for a cell type is √(𝒫_Tot N_C)
33:                 𝒱_Best ← 𝒱^A + 𝒱^B
34:                 BestScheme ← (s^𝔸, s^𝔹, λ^𝔸, λ^𝔹)
35:                 (F̃_Opt 𝔸, F̃_Opt 𝔹) ← (F̃_𝔸, F̃_𝔹)
36:             end if
37:         end if
38:     end for
39:     return BestScheme, F̃_Opt 𝔸, F̃_Opt 𝔹, 𝒱_Best
40: end function
```

where $\tilde{\mathrm{F}}_{\mathrm{Opt}}(\mathbf{S}, \lambda)$ is given by Eq. 43, and where $\mathcal{V}(\tilde{\mathrm{F}}, \mathrm{S})$ is defined as the total variance explained from that mode in the objective function (Eq. 5):

$$\mathcal{V}(\tilde{\mathrm{F}}, \mathrm{S}) = \frac{\mathrm{S}^2|\tilde{\mathrm{F}}|^2}{\sigma_\eta^2 + \mathrm{S}|\tilde{\mathrm{F}}|^2}$$

The actual algorithm implemented contains several optimizations for the sake of speeding up the calculation and recycling work, but here we present the "brute force" algorithm for simplicity. We are currently implementing a more sophisticated algorithm that will not necessitate a full grid search (in preparation).

Note that we optimized the two types for the case of one dimension in space and one dimension in time, and we do not consider the contribution of the noise to the mean firing rate to avoid a dependence on temporal frequency cutoffs. We also matched experimental data of Fig. 3F by optimizing cell types with *two* spatial dimensions and time (data not shown, code available online).

# F Proportion of midget cells as a function of total density in the human retina

To estimate the ratio of midget/parasol cells with respect to total density in the primate retina, we first use the fitted equations of [23] relating the dendritic field size of midget and parasol cells to eccentricty in the human retina:

$$S_M = 8.64 * E^{1.04} \tag{83}$$
$$S_P = 70.2 * E^{0.65} \tag{84}$$

where $S_M$ and $S_P$ are the diameters of the dendritic field of midget and parasol cells respectively and where $E$ is the eccentricity given in $mm$ from the fovea. Then we estimate the corresponding density of midget and parasol cells across retinal eccentricities by using the known overlap between dendritic trees of parasol cells and of midget cells. This overlap has been shown to be constant across the retina, and approximately equal to 1 for midget cells and to 3 for parasol cells [19, 20]. This yields the densities for midget ($D_M$) and parasol ($D_P$) cells:

$$D_M \propto \frac{1}{S_M^2} \tag{85}$$
$$D_P \propto \frac{3}{S_P^2} \tag{86}$$

From these equations we can estimate the fraction of midget cells for any total density of cells $D_M + D_P$. We plot this curve for eccentricities ranging from 0 mm to 20 mm of the fovea.

# G Neural network simulation

We trained a one-hidden-layer convolutional autoencoder to reconstruct pink noise movies, 10 frames in length with 24x24 photoreceptors, generated according to the spectrum of natural movies. Encoding and decoding layers were implemented as 3D convolutional filters large enough to span the entire movie. A ReLU nonlinearity was applied to the hidden layer and an $\ell_1$ constraint was imposed on hidden layer activations as in the theory. The generated pink noise dataset contained 50,000 images.

We added a constant noise magnitude of 0.1 to the output of the encoding layer neurons. $\ell_1$ penalty coefficients of $\lambda = 10^{\alpha+\beta}$ were chosen, with $\alpha \in \{0.5, 0, -0.5\}$ and $\beta$ chosen such that $\lambda(\alpha = 0) = 4$. For a given $\ell_1$ activation budget, grid search over strides that evenly divide the image width (24 photoreceptors) was used to determine the optimal number of cells (i.e. number of neurons in the hidden layer) for each type (i.e. convolutional channels), and gradient descent was used to optimize receptive fields given these hyperparameters. Each cell type allocation was trained from scratch twice, and plots of performance vs. cell type ratio were obtained by computing the convex lower-bounding hull of all the trials. Cell types were grouped in pairs with an equal number of cells. These were implemented as convolutional channels with strides chosen to yield the desired number of cells. For computational tractability, convolutional filters were given a spatial dilation factor of two. The networks were trained using the ADAM optimizer for two epochs, which was found to yield near-convergence. To bias the network toward respecting causality, convolutional padding was added in asymmetric fashion to the temporal dimension, 70% before the start of the 10-frame movie and 20% after the end, with the total padding sufficient to maintain constant dimensionality in each layer. Thus 80% of encoding and decoding weights connected inputs of earlier time coordinates to outputs of equal or later time coordinates (70% to later coordinates, 20% to equal). We found that learned RFs often contained seemingly extraneous high spatial frequency components; we hypothesize that this phenomenon arises from the fact that high frequency modes have low power and hence, though their presence in the learned filters may not be optimal, they contribute very little to the $\ell_1$ firing rate penalty. We show real and Fourier space RFs with Gaussian blur for clarity of visualization.

Note that to make our optimization over four cell types tractable, the grid search over number of neurons per cell type was performed with the restriction that cell types come in pairs with equal numbers of neurons. In fact, primate retinas are thought to contain more OFF ganglion cells than ON cells [23]. This asymmetry has been shown to be an efficient encoding strategy due to the skewness of natural scene statistics, which are not present in our simplified model [13]. Thus, a priori, we had no reason to expect asymmetric ON/OFF population sizes to be beneficial. Nevertheless, we explored the question of whether some asymmetric allocation of cells might be more efficient. We tested the effect of perturbing the cell allocation slightly from its optimal arrangement as determined by the paired-cell-type optimization. Specifically, given the optimal allocation for four cell types of $(N_A, N_A, N_B, N_B)$ cells, we optimized the model for allocations of the form $(N_A + c, N_A - c, N_B, N_B)$,

$(N_A, N_A, N_B + c, N_B - c)$, and $(N_A + c, N_A - c, N_B + c, N_B - c)$ for small values of $c$ = 1,2,3. Thus we perturbed the symmetry of the allocation while preserving the total number of cells. We found that such perturbations reduced the reconstruction performance compared to the optimal model, suggesting that a solution with equal-sized ON and OFF populations of midget and parasol cells is indeed efficient, at least for symmetric Gaussian image statistics.

## H   Effect of firing rate budget on cell type properties

Here we show quantifications of the effect of increasing firing rate budget (or equivalently, decreasing the target reconstruction error). The optimal ratio of midget to parasol cells grows more asymmetric as firing rate penalty decreases in the theory (Fig. 6A) and our neural network simulation (Fig. 6B). Moreover, the benefit of two cell types grows more pronounced as the firing rate budget increases, but is significant across a wide range of parameter settings (Fig. 6C).

Figure 6: A. Firing rate budget as a function of the fraction of midget cells, for different particular fixed acceptable reconstruction errors (from top to bottom: 25%, 10%, 1%) . Note that the optimal cell type ratio shifts to be increasingly asymmetric as the required reconstruction accuracy increases. B. Optimization loss in the neural network simulation as a function of the fraction of midget cells, for different values of $\lambda$ (from top to bottom, $\log(\lambda) = \beta + 1, \log(\lambda) = \beta + 0.5, \log(\lambda) = \beta$, where $\beta$ is a fixed constant) . Note that the optimal cell type ratio shifts to be increasingly asymmetric as the required reconstruction accuracy increases. C. Top: Reconstruction error as a function of firing rate budget, for optimal one cell type and two cell-type solutions. Error decreases as budget increases, two cell types outperform one, and the disparity grows with the firing rate budget. Middle: The relative reduction in error at varying firing rate budgets for the optimal two cell type solution as compared to the optimal one cell type solution. Bottom: The relative reduction in firing rate required to achieve varying target reconstruction errors for the optimal two cell type solution as compared to the optimal one cell type solution. Note that the benefit is significant across a wide range of target reconstruction errors, and it is especially high when high accuracy is required.

## I   RFs of real midget and parasol cells

Receptive fields (RFs) of real midget and parasol cells were obtained by measuring ganglion cell responses of an explanted macaque retina with a multi-electrode array [43], in response to a randomly flickering black&white checkerboard stimulus. The checker size was 22.5 micron on the retina; the frame rate was 60 Hz. After isolating cells with a spike sorting procedure [44] and identifying ON and OFF midget and parasol cells [43],

the spike-triggered-average (STA) of each cell was computed, and an average RF was obtained for each cell type by aligning each cell RF of a given type with respect to their center (defined as the max intensity pixel of the RF of each cell), and then applying a smoothing function on the average RF (default bicubic function of MATLAB - weighted average of pixels in the nearest 4-by-4 neighborhood).

We then performed the same analyses on these RFs that we did for our simulated RFs in fig 4.

Full spatio-temporal RFs for simulated and real ganglion cells are shown in Fig. 7.

Figure 7: A. Receptive fields for four cell types across time, as computed by the neural network simulation (same settings as in Figure 4). B: Spike-triggered averages for real macaque cells. One frame corresponds to a temporal bin of 16.7ms.