[Reviews · NeurIPS 2018]

Reviewer 1



The emergence of multiple retinal cell types through efficient coding of natural movies This papers proposes to study possible distributions of retinal cell types. It builds upon existing literature on four cell types and evaluate how two different configurations of cells could optimally encode natural movies with a unique cell firing rate budget. It draws analogies with optimal configurations and configuration existing in primates. Models are mention to capture 90% of all ganglion cells. The paper tackle an interesting question on wether retinal cell configurations are mathematically optimized for spatio-temporal stimuli. As a disclaimer, I am not an expert on retinal cells. The paper refers to important literature on the matter and focus on mathematical models of cell distributions. However, little information is provided on the type of target visual stimuli. Naively, one may think that different stimuli should be captured differently across varying cell distributions, and one may question why choosing to study one configuration of cells that optimized with one particular stimuli. On the model itself, the firing-rate budget is clear. However, the link with neural networks is less clear. Is the neural net used here historically to truly mimic a biological neural network? Could other models be used, is there a particular architecture, number of layers, to follow in such case? The paper evaluates a the performance of a mixture of 2 types of low/high spatial/temporal cells, and compares it with a configuration of one unique cell types. Why two types and not more configurations? Minor, what is motivation of scaling (eq 3) - notation of footnote (2) can be confusing (squared?) - footnote (4): empirically? - ref [6]? - there are 8 appendices, pushing the length up for this submission The authors' feedback proposed clarifications on major points, including on distributions of stimuli (statistics) vs specific stimuli - overall indicating importance of this work on receptive field configurations.

Reviewer 2



The authors demonstrate that, based on the spatiotemporal statistics of natural movies, it is advantageous (from an energy minimizing perspective) to maintain a retina with more than one convolutional cell type. A key advance is that, whereas prior work in this domain has generally focused on the *spatial* statistics of neural images, here the authors make critical use of the temporal properties of natural movies. This allows them to construct a theoretical justification for several types of retinal ganglion cell. Overall I think this is a well-written and interesting paper.

Reviewer 3



In this paper, the authors present a theoretical treatment of the question why there are multiple cell types in the retina. While efficient coding theories have provided some insight into retinal structure in the past, little work on explaining cell type diversity has been done, with the notable exception of understanding ON- and OFF-pathway splitting. Given recent interest in cell type diversity in the retina, the contribution is timely and targets a fundamental question in systems neuroscience, and the fit to data from midget and parasol cells of the primate is impressive. The linear model studied in sections 3 and 4 provides analytical insights clearly going beyond previous work. Just a few comments here: - lines 109-125: These contain the first main result of the paper and seem correct, but could be argued more clearly. I found the argument hard to follow. - lines 142 f: suddently, the authors use the term "mode" and my impression is that this term is never defined anywhere. Please clarify and explain what you mean. - Fig. 2D/E: Improve the contrast of the line - Fig. 2A-C: I find it surprising that the optimization procedure returns filters with sharp boundaries such as shown in this Figure. How does this come about? Can the RFs be visualized in a more conventional manner as well? In addition, the authors use a 1-layer autoencoder with relus as a slighly non-linear autoencoder. This part feels not complete as the rest of the paper, since RGCs are obviously not the first stage of visual processing, but the third, so a 3-layer autoencoder with relus at each stage would be the more reasonable choice. Overall, it would be nice to see the authors refer to other work than done in the primate as well (reviewed e.g. by Sanes and Masland) and it would be nice to hear the authors thoughts how the primate compares to the mouse in terms of ecological demands.